# Multi-Strategy Improved Cantaloupe Pest Detection Algorithm

**DOI:** 10.3390/insects16121201

**Published:** 2025-11-26

**Authors:** Hongyan Zou, Zishuo Weng, Maocheng Zhao, Xuesong Jiang

**Affiliations:** College of Mechanical and Electronic Engineering, Nanjing Forestry University, Nanjing 210037, China; zouhy@njfu.edu.cn (H.Z.); zsweng2025@163.com (Z.W.)

**Keywords:** cantaloupe pests, crop protection, YOLOv12, object detection, deep learning

## Abstract

The cultivation history of honeydew melons in China spans nearly two millennia, yet insect pests frequently inflict economic losses upon them. Traditional manual inspection methods are inefficient and costly. With the advancement of convolutional neural networks, deep learning has also found applications in the field of pest detection. This paper proposes a lightweight melon pest detection algorithm designed to enhance detection accuracy while maintaining a low number of model parameters and computational requirements. Experimental results demonstrate that the proposed method outperforms mainstream algorithms, achieving higher detection accuracy and lower parameter complexity across datasets of melon, rice, and corn pests. This method provides some guidance for pest control.

## 1. Introduction

Pests are one of the main factors causing major economic losses in crops [1]. Under the current dual demands of agricultural production and environmental protection, it is particularly important to adopt efficient and green pest detection methods [2]. Traditional pest monitoring usually relies on manual visual identification of pest species and populations. However, this method is inefficient, costly, subjective, and limited in scope. It cannot obtain real-time information on the dynamics of pest populations and is prone to untimely prevention and control, resulting in economic losses [3,4].

With the application of convolutional neural networks in image recognition, multi-target recognition in complex scenarios has become increasingly feasible. How to utilize this technology to study plant disease and pest recognition has become a highly concerning topic among researchers [5,6]. In 2021, Zhao et al. [7] designed PestNet, a pest classification model based on saliency detection, by simulating the main process of object recognition by the human visual nervous system. The model mainly consists of the target localization module OPM and the multi-feature fusion module MFFM. OPM integrates shallow detail information and deep spatial information of pest images through a U-shaped network structure, initially delineates the salient regions, and outputs the spatial semantic features. MFFM weakens the background information and increases the detailed features through a bilinear pooling operation of spatial semantic features and abstract semantic features. Guo et al. [8] constructed an attention mechanism based on salient graph analysis to effectively narrow down the range of traps to be detected. This makes the network more focused on dealing with the pest region, which in turn mitigates the problem of misdetection and ultimately improves the detection accuracy. Tang et al. [9] proposed the Pest-YOLO network model, a real-time agricultural pest detection method based on an improved convolutional neural network (CNN) and YOLOv4. The model introduces the SE attention mechanism module and designs a cross-stage multi-feature fusion method to improve the structure of feature pyramid networks and path aggregation networks, thus enhancing the feature expression ability. Ramazi et al. [10] employed machine learning models to study short-term and medium-term predictions of mountain pine beetle outbreaks in Canadian forests. They found that the Gradient Boosting Machine (GBM) performed best for short-term forecasts, while ensemble models demonstrated superior performance for medium-to-long-term projections. This methodology provides reliable support for forest pest monitoring and control decision-making. In 2022, Sun et al. [11] proposed a forestry pest detection method based on the attention model and lightweight YOLOv4. It contains three improvements: replacing the backbone network, introducing the CBAM attention mechanism, and introducing the Focal Loss optimization loss function. This achieved 93.7% mAP on a dataset containing seven forestry pests. Dong et al. [12] proposed a multi-category pest detection network, MCPD-net, to address the problems of great difficulty in detecting small targets and the similarity in appearance of certain pests. It achieved 67.3% mAP and 89.3% average recall (AR) in experiments on the multi-category pest dataset MPD2021. Bjerge et al. [13] proposed an automated real-time insect monitoring system based on computer vision and deep learning. By integrating the YOLOv3 model with the Insect Classification and Tracking (ICT) algorithm, end-to-end processing for insect detection, species classification, and individual tracking has been achieved. In 2023, Zhao et al. [14] proposed a pest identification method based on improved YOLOv7 for the complex environment of farmland. By combining the CSP Bottleneck with the shift-window Transformer self-attention mechanism based on the shift-window, increasing the detection branch, introducing the CBAM attention mechanism, and adding the Focal EIoU Loss Function, this four-point improvement made the mAP of the model reach 88.2%. Wang et al. [15] proposed a single-stage unanchored detection network, OSAF-Net, with robust performance against the challenges of distinguishing between similarly shaped pests and multi-scale pests, leading to a large number of false-negative detections. It achieved good detection results on both CropPest24 and MPD2018 datasets. Duan et al. [16] introduced the SENet module and Soft-NMS algorithm in YOLOv4 to solve the problem that corn pests are not easy to recognize and improve the detection accuracy. Choiński et al. [17] developed a method using deep learning to automatically detect insects in photographs, enabling automatic feature extraction from raw images for insect detection and counting. In image tests conducted in Poland, Germany, Norway, and other locations, the method achieved a precision of 0.819 and a recall of 0.826. Badgujar et al. [18] proposed a real-time detection and identification system for storage pests based on deep learning (YOLO series models), providing an end-to-end framework for automated and real-time insect detection and identification in stored product environments. Salamut et al. [19] achieved automated detection of cherry fruit flies in sticky traps using deep learning. Among the five models employed, the optimal average detection accuracy reached 0.9, addressing the inefficiency inherent in traditional monitoring methods. Bjerge et al. [20] constructed a large-scale image dataset of insect taxa. Their YOLOv5 model addressed the challenge of detecting small insects against complex vegetation backgrounds, advancing the field of insect monitoring. In 2024, Zhou et al. [21] proposed a small-objective multi-category farmland pest target detection algorithm based on YOLOv5 improvement to address the problem of inconspicuous characteristics of farmland pests and the predominance of small pests. It achieved a mAP of 79.4% on a publicly available dataset containing 28 categories of farmland pests. Li et al. [22] proposed a lightweight, location-aware fast R-CNN (LLA-RCNN) method. To reduce the computational effort, the model uses MobileNetV3 to replace the original backbone and introduces the CA Attention Mechanism Module for augmenting the location information. In addition, the generalized intersection and union set (GIoU) loss function and region of interest alignment (RoI Align) technique are used to improve the pest detection accuracy. Elmagzoub et al. [23] proposed a rice pest identification method integrating deep learning feature extraction and feature optimization. The ResNet50 model, combined with feature vectors extracted via LR and PCA, achieved an accuracy rate of 99.28% in identifying rice insects. Hacinas et al. [24] developed an edge computing application for low-end mobile devices that enables automated counting of cocoa pod borers (CPB) on sticky traps using an optimized YOLOv8 model. This study provides a feasible solution for low-cost pest monitoring. In 2025, Xiong et al. [25] proposed an improved detection model, QMDF-YOLO11, based on YOLO11 for farmland pests in complex scenarios. It achieved a mAP of 94.57% on the RicePests dataset, effectively solving the problem of low accuracy of pest detection in complex backgrounds and small target scenarios. Zhang et al. [3] proposed a lightweight farmland pest detection algorithm based on YOLOv7 to address the problem of a large number of parameters and computation in the current pest detection algorithm model. It achieved 72.1% detection accuracy for farmland pests, while the computation and number of parameters were kept at a low level. Liu et al. [26] proposed an end-to-end pest detection method based on feature representation compensation (FRC) and region-like shape self-attention (RPSA). They designed a CSWin-based FRC module to compensate for the loss of feature information due to the downsampling process and proposed an RPSA-based Transformer encoder to capture global information and enhance the local information of the feature map. Rajeswaran et al. [27] proposed a method for identifying live insects in agricultural scenes using motion analysis of consecutive video frames, and compared five deep learning object detection models. Experiments demonstrate that the SSD_MobileNet_V2 model delivers optimal performance, providing a lightweight solution for precision agricultural insect management. Kargar et al. [28] proposed SemiY-Net, a compact deep learning model for insect segmentation and image-dependent tasks, achieving ideal pest detection and counting performance on MCU-based circuit boards. This research provides edge computing solutions for precision agriculture. Ong et al. [29] investigated how the color of sticky traps and the imaging equipment used affect the effectiveness of deep learning in automatically identifying insects on sticky traps. It is proposed that to build a stable and reliable automated insect monitoring system, simultaneous optimization is required in both trap color selection and deep learning network architecture design.

Cantaloupe leaves are susceptible to pests during the growth process. Through regular, accurate identification and monitoring of cantaloupe leaf pests in the field, we are able to detect the traces of pests earlier. This allows us to implement prevention and control to prevent outbreaks of pest populations, reduce the use of chemical pesticides, and reduce the damage to cantaloupe fruits. In terms of the current state of research, machine vision has received extensive attention in the field of plant science [30]. This study proposes a detection method for leaf pests on honeydew melon plants using YOLOv12 as the baseline model. It addresses challenges such as dense pest distribution, minute body size, and inconsistent dimensions in melon fields. The approach aims to enhance detection performance across diverse scenarios while maintaining model weight optimization. Additionally, this study validates the effectiveness of the improved model in detecting pests of other crops through generalization experiments, providing technical guidance for the control of agricultural pests.

The main contributions of this study are as follows:An improved YOLOv12 algorithm is proposed, specifically optimized to address the challenges posed by melon pests, such as their minute size and inconsistent dimensions. This enhancement achieves higher detection accuracy while maintaining a low number of parameters and computational requirements. Additionally, the refined model effectively detects pests across various crops, demonstrating excellent adaptability.The C3k2-B module is designed by introducing an adaptive residual block (ARBlock) to replace the original block module in the C3k2 architecture. This transformation creates a lightweight and efficient feature extraction module, enabling the model to capture features across different receptive fields. Furthermore, the fusion layer between the Neck and Head networks is modified to enhance the model’s utilization of shallow-layer features.

## 2. Materials and Methods

### 2.1. Data Sources and Preprocessing

The dataset used in this study was the Melon Cantaloupe Pest dataset from the Robotflow website, which contained complex images of five common cantaloupe leaf pests under natural conditions. The five pests were Aphids, Leafminers, Moths, Red-Melon Beetles, and Whiteflies. Due to the special habits of leafminers, the dataset images were not of the pest itself, but of the zigzagging white trails it left on the leaves. An example diagram of the five categories of pests is shown in Figure 1.

In the data preprocessing stage, samples with serious ambiguities, distortions, and missing labeling information were first removed to ensure the quality of the dataset. Then in order to avoid data leakage (Images in the expanded part of the dataset that were similar to those in the training set were partitioned into the validation set), the pest images of each category were first divided into the training set, validation set, and test set in the ratio of 8:1:1. To mitigate classifier bias and enhance dataset diversity, data augmentation techniques involving horizontal flipping and vertical flipping (each applied with a 50% probability) were employed to augment the training images for aphids and leafminers, which had relatively smaller datasets. The aphids training dataset expanded from 952 to 1904 images, while the leafminers dataset grew from 1100 to 2200 images. After the above processing, the final constructed dataset had a total of 11,949 images, including 9967 images in the training set, 988 images in the validation set, and 994 images in the test set. Examples of some of the excluded problem images are shown in Figure 2. The label information for the training and validation sets is shown in Figure 3. The labels and number of images for each type of data were shown in Table 1.

### 2.2. YOLOv12 Network Improvements

In order to enhance the detection performance of YOLOv12 in the cantaloupe leaf pest task, four improvements were proposed in this study: (1) Add the EMA attention mechanism module after the Conv and A2C2f layers at the end of the backbone network, respectively, to enhance the model’s ability to capture features of small targets; (2) Change the fusion strategy of the Concat layer and the Detect layer of the detection head in the Neck network Neck to construct a smoother feature pyramid, enhance the ability of the model to capture features of pests with different scales, and improve the detection accuracy; (3) Introduce the WIoU v3 loss function to reduce the harmful gradient of low-quality samples and focus on the anchor frames of ordinary quality, thus improving the overall performance of the model; (4) Improve the C3k2 module to enhance the detection accuracy while keeping the model lightweight and reducing the computational and parametric quantities of the model. The structure of the improved YOLOv12 network model is shown in Figure 4.

#### 2.2.1. EMA Attention Mechanism

EMA (Efficient Multi-Scale Attention Module) was proposed by Ouyang et al. in 2023 [31], which aims to enhance feature representation in computer vision tasks through cross-spatial learning and a multi-branching structure. It adopts a three-branch parallel architecture, as shown in Figure 5. First, the input feature maps are divided into g subgroups along the channel dimension to avoid information loss from channel dimensionality reduction and to reduce computational complexity, with each group having dimensionality c//g × h × w. The parallel sub-network structure is divided into 1 × 1 branches in the upper half and 3 × 3 branches in the lower half. In the 1 × 1 branch, each grouping will perform one-dimensional global average pooling (X Avg Pool) in the horizontal direction and one-dimensional global average pooling (Y Avg Pool) in the vertical direction. This operation can help the model grasp the distribution of pests across leaves when detecting cantaloupe pests. The features after the pooling layer are spliced (Concat) to capture cross-channel interactions with a 1 × 1 convolution. The 3 × 3 branch utilizes convolutional kernel local receptive fields to capture local details and multi-scale feature representations, such as leaf miner trajectory details, to compensate for the 1 × 1 branch’s lack of local information. EMA enriches feature aggregation by providing cross-space information aggregation methods across different spatial dimensions. Specifically, the 1 × 1 branch output is encoded with global spatial information by 2D global average pooling. In contrast, the output of the 3 × 3 branch is directly converted to the corresponding dimensional shape. The spatial attention map is generated by performing a matrix dot product operation (Matmul) on the outputs of the above parallel processing. Ultimately, the output feature maps within each group are aggregated by a Sigmoid function of the two generated spatial attention weight values, capturing pixel-level pairwise relationships and highlighting the global context of all pixels.

The formulas for one-dimensional global average pooling (X Avg Pool) in the horizontal direction and one-dimensional global average pooling (Y Avg Pool) in the vertical direction are shown in Equations (1) and (2), respectively:(1)$$Z_{c}^{H}\left(H\right)=\frac{1}{W}\sum_{0\leq i\leq W}x_{c}\left(H,i\right)$$

The above equation represents the average pooling along the horizontal dimension (width *W*) for the cth channel of the input feature map X. The output dimension is *C × H × 1*.(2)$$Z_{c}^{W}\left(W\right)=\frac{1}{H}\sum_{0\leq j\leq H}x_{c}\left(j,W\right)$$

The above equation represents the average pooling along the vertical dimension (height *H*) for the cth channel of the input feature map X. The output dimension is *C* × *1* × *W.*

The backbone network of YOLOv12 is the key to feature extraction, and the introduction of the EMA attention mechanism in this part can effectively highlight the characteristics of the target area and suppress the background interference. Therefore, the model can more accurately focus on the key target information in the subsequent processing, improving its ability to detect small targets.

#### 2.2.2. Improved Neck and Head Network Fusion Strategy

In the Neck network of YOLOv12, the Concat module at layer 17 combines features from the Conv layer above it and the A2C2f module at layer 12. In the Head network, the Detect layer fuses features from the 15-layer A2C2f, 18-layer A2C2f, and 21-layer C3k2 modules, respectively. In the original architecture, the Concat layer of the Neck network and the Detect layer of the Head network receive inputs at relatively deep positions. This arrangement tends to cause the gradual attenuation of spatial details in shallow features rich in positional and contextual information after undergoing multiple convolutions and feature transformations. Consequently, the model’s detection capability for small and edge-bound targets is compromised. Not only that, the layers of feature fusion are deeper, the semantic and spatial information at different scales cannot be fully interacted with at an earlier stage, and shallow features play a limited role, which is not conducive to the model’s effectiveness in detecting multi-scale targets. To address the above problems, this study advances the fusion layer between the Concat layer in the Neck network and the Detect layer in the Head network, enabling the shallow features processed by the EMA attention mechanism to participate in the fusion earlier and more directly within the network. This retains more spatial and texture details and improves the utilization of shallow spatial detail information, and the features of different scales can achieve full interaction at an earlier stage. The improvement of the above-optimized fusion strategy enhances the network’s robustness to multi-scale targets and improves the model’s ability to detect small and dense targets. The structure of the Neck and Head network before and after the improvement is shown in Figure 6.

#### 2.2.3. WIoU v3 Loss Function

The leafminers class of targets in this study’s dataset exhibited irregular white tunnels on leaves with complex and significantly different edge profiles than other insect targets. This type of target is prone to inaccurate localization labeling, such as when the labeling box does not completely cover the channel area or contains too much background, which inevitably produces some low-quality samples. This leads to the traditional IoU-type loss function for irregular target regression being prone to limitations, and the training process fails to adequately focus on difficult samples, which affects detection accuracy. In order to solve the above problem, this study introduces the WIoU v3 loss function into the original YOLOv12 model. WIoU (Wise-IoU) [32] is a bounding box regression loss function based on a dynamic nonmonotonic focusing mechanism (FM), which aims to address the impact of low-quality samples in the training data on the generalization performance of the target detection model. The dynamic nonmonotonic focusing mechanism uses “outliers” as an alternative to IoU for quality assessment of anchor frames and provides a sensible strategy for gradient gain assignment. This strategy reduces the competitiveness of high-quality anchor frames while reducing the deleterious gradients generated by low-quality examples, which allows the WIoU to focus on average-quality anchor frames and improve the overall performance of the detector. WIoU v1 constructed an attention-based bounding box loss, to which WIoU v3 attaches a focusing mechanism by constructing a calculation of the gradient gain (focusing coefficient). Dynamic nonmonotonic FM: The outlier β of the anchor frame is expressed as the ratio of $L_{IoU}^{*}$ and $\bar{L_{IoU}}$:(3)$$\beta =\frac{L_{IoU}^{*}}{\bar{L_{IoU}}}\in \left[0\right.,\left.+\infty \right)$$
where $L_{IoU}^{*}$ is the current anchor frame loss, and $\bar{L_{IoU}}$ is an exponential moving average with momentum m. The average of the anchor frame loss is the current anchor frame loss. A small outlier means that the anchor frame is of high quality, and a small gradient gain is assigned to it in order to focus the bounding box regression to an anchor frame of average quality. Assigning a small gradient gain to anchor frames with large outliers will effectively prevent the low-quality examples from generating large, harmful gradients. A nonmonotonic focusing factor γ is constructed using β and applied to the WIoUv1:(4)$$L_{WIoUv3}=\gamma L_{WIoUv1},\gamma =\frac{\beta }{\delta \alpha^{\beta -\delta }}$$(5)$$L_{WIoUv1}=R_{WIoU}L_{IoU},R_{WIoU}\in \left[1\right.,\left.e\right),L_{IoU}\in \left[0,1\right]$$(6)$$R_{WIoU}=\exp\left(\frac{{\left(x-x_{gt}\right)}^{2}+{\left(y-y_{gt}\right)}^{2}}{{\left(W_{g}^{2}+H_{g}^{2}\right)}^{*}}\right)$$(7)$$L_{IoU}=1-IoU$$
where α and δ are hyperparameters that control the mapping of outlier β and gradient gain γ. When δ=β, γ = 1. In this study, the parameters δ = 3 and α = 1.9. x and y are the center coordinates of the prediction box, $x_{gt}$ and $y_{gt}$ are the center coordinates of the true box, $W_{g}$ and $H_{g}$ are the sizes of the smallest closed box, and the superscript * represents the separation of $W_{g}$ and $H_{g}$ from the computational map. $L_{IoU}$ is the IoU loss and $R_{WIoU}$ is the penalty term for WIoU that will significantly amplify the $L_{IoU}$ of the normal mass anchor frame. Since $\bar{L_{IoU}}$ is dynamic, the quality classification criteria of anchor frames are also dynamic, which enables WIoU v3 to make the most appropriate gradient gain allocation strategy according to the current situation. When the outlier degree of the anchor frame satisfies β=C (C is a constant), the anchor frame will obtain the highest gradient gain.

#### 2.2.4. Improvement of the C3k2 Module

The above improvements of adding the EMA attention mechanism and optimizing the fusion strategy bring about a large computational enhancement. In order to balance the improvement of the YOLOv12 model’s detection capability with its light weight, this study improves the C3k2 module. Adaptive Residual Block (ARBlock) is introduced to replace the original block module, which introduces three parallel branches inside the module, using depth-separable convolution with dilation = 1, 3, 5, respectively, to help the model to obtain the features of different receptive fields, and to improve the ability of multi-scale information fusion, as well as to reduce the number of computation and parameter count of the model. At the same time, it reduces the amount of model computation and the number of parameters. The ECA (Efficient Channel Attention module) [33] is also introduced, which not only enhances the model’s detection of small targets but also keeps the model lightweight.

The improved C3k2 module is named C3k2-B, and the structure is shown in Figure 7. The input features first pass through the transition module, and then enter two branches after splitting. The left branch is directly input to the feature fusion layer (concatenation) after 1 × 1 convolution, and the right branch is also input to n adaptive residual blocks (ARBlock) after 1 × 1 convolution to enhance the feature expression ability, and finally input to the feature fusion layer (concatenation). In ARBlock, the input features are first subjected to 3 × 3 depth-separable convolution, BatchNorm (batch normalization), and SiLU (Sigmoid Linear Unit) activation function operations, and then input to three parallel depth-separable convolution branches with dilation rates set to 1, 3, and 5, respectively. This partially expands the receptive field without increasing the number of parameters, and convolution operations with different dilation rates can capture feature information at different scales, realizing the capture of pest features at different scales. The branch output is then concatenated along the channel dimensions to fuse multi-scale features and enhance the model’s ability to detect pests of different sizes. This is followed by a BatchNorm (batch normalization) operation and the introduction of the ECA (Efficient Channel Attention) attention mechanism to enhance the network’s feature representation ability without incurring high computational costs. Finally, the module employs adaptive gating and residual connections, with the formula for outputting feature maps as follows:(8)$$x_{out}=gate\left(x_{residual}\right)\cdot x_{conv}+\left(1-gate\left(x_{residual}\right)\right)\cdot x_{residual}$$

Here, $x_{out}$ represents the final output feature map, $x_{conv}$ denotes the feature map after the convolution operation, $x_{residual}$ is the input feature map for the residual connection, and $gate\left(x_{residual}\right)$ serves as the gating coefficient, outputting a weight value between 0 and 1. In summary, the Adaptive Gate operates by generating gating weights through 1 × 1 convolutions and the Sigmoid function, dynamically adjusting the weights of the convolutional features $x_{conv}$ and residual features $x_{residual}$ based on these gating weights. If $gate\left(x_{residual}\right)$ is closer to 1, the output is primarily determined by the convolutional feature map $x_{conv}$. If $gate\left(x_{residual}\right)$ is closer to 0, the output is primarily determined by the residual feature map $x_{residual}$.

## 3. Results

### 3.1. Experimental Environment and Configuration

The research in this paper is carried out under a 64-bit Window11 operating system with NVIDIA GeForce RTX 4060 as the graphics card model and 8GB of GPU memory. The deep learning framework is Pytorch 2.6.0, the CUDA version is 11.8, and the Python version of the runtime environment is 3.9. The parameter configurations in the experiment are shown in Table 2.

### 3.2. Model Performance Evaluation Metrics

In order to comprehensively evaluate the performance of the improved YOLOv12 model in the cantaloupe pest detection task, this paper uses the *P* (Precision), the *R* (Recall), the *mAP50* (mean Average Precision, IoU = 0.5), *mAP50-95* (mean Average Precision, IoU = 0.5:0.95), the F1-Score (F_1_), the number of model parameters (Parameters), and the number of model floating-point calculations (GFLOPs) as evaluation metrics. The *P* represents the proportion of samples predicted by the model to be in the positive class that are actually in the positive class, and the *R* represents the proportion of samples in the dataset that are actually in the positive class that are correctly detected by the model. The formulas for *P* and *R* are as follows, respectively:(9)$$P=\frac{TP}{TP+FP}\times 100\%$$(10)$$R=\frac{TP}{TP+FN}\times 100\%$$
where *TP* stands for true cases, i.e., the number of positive class samples correctly detected by the model, *FP* is false positive cases, i.e., the number of samples incorrectly predicted by the model to be in the positive class, and *FN* is false negative cases, i.e., the number of samples of the positive class that were missed by the model. F_1_ is the weighted harmonic mean of *P* and *R*, calculated as follows:(11)$$F_{1}=2\times \frac{P\times R}{P+R}\times 100\%$$

The *mAP* is one of the most central evaluation metrics in the target detection task, which is used to comprehensively measure the detection performance of the model for different categories and different confidence levels of prediction results, and its formula is as follows:(12)$$AP=\int_{0}^{1}P\left(r\right)dr$$(13)$$mAP=\frac{1}{N}\sum_{1}^{N}AP_{i}$$
where *N* is the number of categories of detected targets, *N* = 5 in this study, $P_{i}$ represents the average precision of the ith category. *AP* represents the average precision, and *mAP* is the average of the *AP* values across all object categories; that is, *AP* is computed for each category, and the mean of these *AP* values yields the *mAP*.

### 3.3. Ablation Experiments

To verify the effectiveness of each improvement module, this paper conducts ablation experiments based on the YOLOv12 model as well as the above evaluation indices under the same experimental environment and configuration. The experimental results are shown in Table 3. After the introduction of the EMA attention mechanism module in the backbone alone, the mAP50, P, and mAP50-95 metrics have all risen slightly. This indicates that the EMA attention mechanism improves the model’s ability to capture key target features, eliminate background interference, and enhance detection accuracy. After introducing the fusion strategy alone, the four indicators of mAP50, P, R, and mAP50-95 improved by 1.2, 0.35, 1.52, and 1.61 percentage points, respectively. The slight decrease in parameters indicates that the fusion strategy helps the model integrate features at different levels more effectively. It also helps the model adapt to targets of different sizes by adjusting the hierarchical structure of the feature pyramid and simplifying the fusion module. After introducing the C3k2-B module alone, the four indexes were improved by 0.66, 0.29, 1.05, and 1.03 percentage points. The number of parameters and the computational volume were reduced to 2.42M and 5.9 GFLOPs, respectively, which indicated that the improved C3k2 module could still effectively extract pest features and improve the detection capability of the model while ensuring that the model is lightweight. After introducing the WIoU v3 loss function alone, the mAP50, R, and mAP50-95 metrics are all improved to different degrees, while the number of parameters and computation remain unchanged. However, P slightly decreases, which is probably due to the fact that the dynamic focusing mechanism of WIoU v3 adjusts the training weights of the different anchor frames, which breaks the balance between the model’s original classification and regression. This leads to the occurrence of classification accuracy and, in turn, a small fluctuation in classification accuracy. The role of WIoU v3 will become more evident when synergized with other modules.

Further, with the simultaneous introduction of the EMA attentional mechanism and the fusion strategy, the four metrics of mAP50, P, R, and mAP50-95 of Model 6 were improved by 2.37, 1.99, 1.87, and 2.37 percentage points, respectively, when compared to Model 1. The combined introduction of the two improvements was found to be more effective. This indicates that the synergy between the EMA attention mechanism and the fusion strategy is good. Combining the two can significantly enhance the model’s ability to detect targets at different scales, such that the model can better utilize the different levels of features, which has an obvious effect on the overall performance enhancement. Subsequently, the WIoU v3 loss function was introduced, resulting in improvements in both mAP50 and R for the model. Finally, by adding the C3k2-B module to Model 7, Model 8 achieved an mAP50 of 85.06%, P of 86.76%, R of 79.94%, and mAP50-95 of 54.15%. Compared to baseline Model 1, the four metrics improved by 3.48%, 3.17%, 2.5%, and 3.45%, respectively. Moreover, the model parameters were reduced, and computational complexity did not increase significantly. Not only that, it can be seen from Models 2, 3, 4, 5, and Model 8 that the synergistic improvement effect of the four improvement strategies is better than the superposition of the improvement effect of each improvement strategy. Therefore, in summary, the four improvement strategies proposed in this study work closely and synergistically with each other, which can not only effectively enhance the model’s detection performance of cantaloupe pests and improve the detection accuracy, but also optimize the model structure, make the model more lightweight, and have good comprehensive performance.

Comparison plots of mAP, P, R, and mAP50-95 results for models 1, 2, 6, 7, and 8 in the ablation experiments are shown in Figure 8.

The values of the average detection accuracy mAP50 for each type of pest after each improvement point was added individually are shown in Table 4. As can be seen from the table, all four improvement strategies contribute significantly to the improvement of aphid detection accuracy. The introduction of the WIoU v3 loss function and fusion strategy significantly improves the recognition accuracy of leafminers. The accuracy of moth and red-melon beetle detection was already very good, and the results of the model before and after the improvement were not very different. The introduction of the fusion strategy has greatly improved the accuracy of whitefly detection. Thus, each improvement strategy plays a different role and has a different degree of effectiveness in detecting each type of pest. The boosted values of the improved YOLOv12 model for each type of pest AP are shown in Figure 9.

### 3.4. Performance Comparison Between C3k2 and C3k2-B Modules

The 11th, 14th, 17th, and 20th layers of the model were selected to be indexed to draw the heat maps, and the detection results and heat maps are shown in Figure 10. From the detection results, compared with the C3k2 module, the introduction of the C3k2-B module improved the detection of pests at different scales to a certain extent. At the same time, the module improved the problems of misdetection and duplicate frames (marked by the red circle in the Figure 10). From the heat maps, after the introduction of the C3k2-B module, the pest contours are more obvious, and the model is more capable of capturing detailed features such as antennae and legs. In addition, the introduction of the C3k2-B module decreases both the number of parameters and the computational effort of the model, achieving an FPS (frames per second) of 70.92 on the test set, which is only about 0.5 lower than the 71.43 of the C3k2 model. The above results show that the C3k2-B module better maintains the lightweight nature of the model while improving the model detection capability.

### 3.5. Comparison Experiments with Other Models

As shown in Table 5, this study compares the improved YOLOv12 model with other mainstream target detection algorithms. The experimental environments and parameter configurations are the same, and none of them add pre-training weights. As can be seen from the table, the improved YOLOv12 network model in this paper outperforms all other YOLO series comparison models in the four metrics of mAP50, P, mAP50-95, and F_1_. Although the R is slightly lower than that of the YOLOv5s model, it performs better on the remaining three metrics, especially on the mAP50-95 metric, where the improved YOLOv12 model outperforms the YOLOv5s by 8.21% and is less computationally intensive. Not only that, the improved model parameter count is only 2.43M, which is lower than all comparison networks except YOLOv5n. In addition, the improved YOLOv12 model outperforms the second-best algorithm, YOLOv11, by 1.95, 2.53, 1, 1.37, and 1.76 percentage points in mAP50, P, R, mAP50-95, and F_1_, respectively. It also features fewer parameters and requires only 4.69% more computational effort.

RT-DETR (Real-Time Detection Transformer) [34] is a relatively new model in the DETR family of algorithms, which is an innovative model architecture that combines the Transformer and real-time target detection. It aims to solve the problem of the trade-off between speed and accuracy in the existing target detection models by introducing an efficient Transformer module and an optimized detection head, which improves the real-time performance and accuracy of the model. Compared to the RT-DETR model, the improved YOLOv12 model exhibits an R that is 0.48% lower than the former. However, it outperforms the RT-DETR model by 2.51, 2.08, 3.27, and 1.02 percentage points in the four metrics of mAP50, P, mAP50-95, and F_1_, respectively. Additionally, it features a lower number of parameters and computational complexity. Therefore, in a comprehensive view, compared with other models, the model proposed in this paper achieves improved detection accuracy while maintaining lower computational and parametric quantities, and exhibits superior lightweighting and detection performance.

Figure 11 shows the confusion matrix for the YOLOv12 model, the suboptimal model YOLO11, and the improved YOLOv12 model.

Figure 12 shows the validation loss curves for YOLOv12, the suboptimal model YOLOv11, and the improved YOLOv12. As shown in the Figure, the loss of all three models gradually decreases with increasing training iterations, eventually converging to a stable value. Notably, the improved YOLOv12 model converges to a lower value.

### 3.6. Comparative Experiments on Adding Different Attention Mechanisms

In order to verify the effectiveness of the EMA attention mechanism module in the cantaloupe pest classification task after synergizing with other modules, different attention mechanism modules are introduced for comparison experiments, and the experimental results are shown in Table 6.

As can be seen from the table, the introduction of the EMA attention mechanism in the improvement strategy outperforms several other contrasting attention mechanisms in the three metrics of mAP50, P, and mAP50-95, and there is little difference in the number of parameters and computational effort of the several models. With the introduction of CA, SimAM, and SE attention mechanisms, R is all slightly improved over the original improved network, but the other three metrics, mAP50, P, and mAP50-95, are lower than the original improved network. The remaining four metrics for the three sets of attention mechanism comparison tests were lower than those for the original improved network. Therefore, combining the indicators, the effectiveness of the EMA attention mechanism when synergized with the other modules is superior to the other contrasting attention mechanisms.

Additionally, it is worth noting that the R improved slightly after introducing the CA, SimAM, and SE attention mechanisms. The SE (Squeeze and Excitation) attention mechanism obtains channel descriptions through global average pooling and uses two fully connected layers to generate channel recalibration weights, significantly enhancing inter-channel dependencies. SE enhances the overall response of the channel, amplifying weak targets such as whiteflies. However, due to the lack of fine-grained spatial selection, background textures may also be amplified simultaneously. Compared to EMA, this approach tends to yield a slight increase in R but a slight decrease in P and mAP. The CA (Coordinate Attention) mechanism decomposes the 2D global pooling operator into two 1D directional encodings along the height and width axes. This preserves precise positional information while establishing long-range dependencies, ultimately generating direction-aware and position-sensitive channel attention maps. CA embeds directional position information within the channel attention mechanism, making it more sensitive to targets with directional structures (such as leafminers’ tunnels). This enhances its ability to extract trajectory features from weak targets, thereby improving R. However, its spatial gating remains primarily channel-based, and its suppression of complex backgrounds is less effective than multi-scale pixel-level feature modeling. SimAM (Simple Attention Module) calculates 3D attention weights for each spatial position-channel dimension based on the principle of minimizing the neural energy function, without requiring additional parameters. SimAM employs energy-based pointwise weighting, which globally enhances salient responses (such as the red coloration of the red-melon beetles), typically resulting in a slight improvement in R. However, due to the lack of explicit multi-scale contextual modeling and cross-spatial interactions, it is challenging to effectively distinguish the boundaries and features of adjacent targets in scenarios where targets such as whiteflies and aphids are clustered. Consequently, its contribution to mAP is relatively limited. EMA is a multi-scale contextual attention mechanism that balances local details with global semantics. It suppresses background noise and focuses on small target features in scenarios with small objects, large-scale variations, and complex backgrounds. This enables the model to more accurately identify pest species and locations, making it better suited for the melon pest detection task. Consequently, it achieves the best performance across three metrics: mAP50, P, and mAP50-95. The R values for CA, SE, and SimAM attention mechanisms are slightly higher than those for EMA, as they enable the model to amplify weak target signals. However, all three suffer from insufficient spatial or multi-scale selectivity, meaning their relatively high R values also come with a higher rate of misclassification. If the deployment scenario prioritizes recall (e.g., early warning systems), the gap in R can be addressed by methods such as category-specific thresholds while maintaining the EMA.

### 3.7. Experiments to Improve Model Generalizability

In order to verify the generalization ability of the improved model in this study, the Pest-dataset and Rice Pests datasets from the Robotflow website are introduced as the datasets for the generalizability experiments. We screened the images and annotations. Both datasets feature high annotation quality, appropriately sized bounding boxes, and no instances of missing annotations or excessively blurred images.

The Pest-dataset contains complex images of six corn pests in their natural environments, with the pest categories being army worm, black cutworm, grub mole cricket, peach borer, and red spider. The dataset contains a total of 3899 images, of which 2730 are in the training set, 779 are in the validation set, and 390 are in the test set. The images are resized from 416 × 416 to 640 × 640. The Rice Pests dataset contains six rice pests: brown planthopper, green leafhopper, leaf folder, rice bug, stem borer, and whorl maggot. The dataset contains a total of 5229 images, of which 3922 are in the training set, 1046 are in the validation set, and 261 are in the test set. The above two datasets have the advantages of rich features, diverse pest postures, and complex collection scenarios, which can effectively evaluate the generalization ability and robustness of the model in the crop pest detection task. A comparison of the basic information for the three datasets, Melon Cantaloupe Pest, Pest-dataset, and Rice Pests, is shown in Table 7.

The generalizability experiment 1 uses the Pest-dataset, and its results are shown in Table 8. Among them, the values of the four metrics mAP50, P, R, and mAP50-95 of the improved model are 94.08%, 94.03%, 89.15% and 66.99%, respectively, which are improved by 5.44, 6.14, 8.12, and 3.5 percentage points compared to the original model. The generalizability experiment 2 uses the Rice Pests Dataset, and its results are shown in Table 9. Among them, the values of the four metrics mAP50, P, R, and mAP50-95 of the improved model are 90.91%, 91.17%, 85.60% and 60.17%, respectively, which are improved by 4.82, 5.43, 4.34, and 5.5 percentage points, respectively, compared to the original model. The above two sets of generalization experiments show that the improved model can effectively adapt to pest detection scenarios of different crops, reflecting its strong generalization ability and applicability.

## 4. Discussion

### 4.1. Model Detection Performance Comparison

In order to show the detection effect of the improved YOLOv12 model more intuitively, representative pest images were selected for testing. The detection effects of the YOLOv12 model before and after the improvement were compared with the suboptimal model YOLO11 in different scenarios, such as pests with different size scales, different backgrounds, diverse morphologies, and the presence of different degrees of occlusion.

#### 4.1.1. Comparison of Pest Detection at Different Scales

In cantaloupe leaves, pests of different sizes are often present, which makes accurate detection of the model difficult. Figure 13 shows the detection results for each model in this case. In the images of groups (a) and (b), aphids are densely distributed, vary in size, and are partially obscured. By refining the fusion strategy of the YOLOv12 model and introducing convolutions with different expansion rates in the C3k2 module, the model’s ability to capture features of pests of different sizes is enhanced. This improves the detection accuracy of pests under these conditions and mitigates the issue of duplicate detections present in the original model to a certain extent. In group (c), the whiteflies are tiny and make up much less than 5% of the entire picture. Compared to the original model and YOLOv11, the improved YOLOv12 model incorporates an EMA attention mechanism, enhancing its detection capability for small objects. Consequently, it achieves higher confidence scores, and the improved model does not exhibit the duplicate detection issues present in the original model.

#### 4.1.2. Comparison of Different Background Pest Detection

In cantaloupe fields, there are often scenes that interfere with pest detection due to lighting, shooting angle, and leaf–root interlacing, such as situations where the color of the background is similar to the pest and contains complex information. Figure 14 shows the detection results for each model in this case. In group (a), the moth’s color is similar to the background, and the improved YOLOv12 model achieves a higher confidence level. In group (b), complex backgrounds such as the ground and leaves contain a large amount of texture and spatial information, and the original model may not be able to blend shallow features well, leading to missed detection. The improved YOLOv12 model introduces a new fusion strategy, which preserves more spatial and textural details, and is able to more accurately extract features such as the shape contour of the red-melon beetles, avoiding leakage of detection due to insufficient feature extraction, and also achieves a higher confidence level. In group (c), which contains complex backgrounds including leaves, roots, stems, and fruits, the number of detection targets is relatively low. In this scenario, the improved YOLOv12 model does not exhibit the duplicate detection issues seen in YOLOv11 and demonstrates higher confidence levels.

#### 4.1.3. Comparison of Detection Results of Different Morphological Targets

Pests on cantaloupe leaves have different morphologies, including shape and posture. Figure 15 illustrates the detection results of different models for targets with different morphologies. The aphids in group (a) are quite different in appearance from those illustrated in Figure 13 above. The red-melon beetle in group (b) is in a more unusual posture, hanging from the leaf at an angle. In contrast, group (c) presents the passage of leafminers with varying morphology. From the detection results, the improved YOLOv12 model achieves a higher confidence level when faced with different samples of pests. In group (b), the YOLO11 model appears to be misdetected. In group (c), the leafminers’ tunnels are not clearly divided into individuals, and it is also difficult to avoid the inclusion of a large amount of background information when labeling these types of samples, and the percentage of background in the labeling box is often much larger, which brings considerable difficulties in detection. The improved YOLOv12 model introduces the WIoU v3 loss function, which helps the model learn the features of the target itself more efficiently and reduces the impact of difficult samples on the model’s generalization performance, thus increasing the accuracy of the detection and improving the leakage problem that occurs in the original model. However, at the same time, the improved model also suffers from redundancy in the detection frames, which is also specifically summarized in Section 4.2.

#### 4.1.4. Comparison of Pest Detection Results Under Different Levels of Shading

Figure 16 demonstrates the detection results of different models for red-melon beetles with different degrees of occlusion from group (a) to (c). As the degree of occlusion gradually deepens, the model is able to extract less and less information about the pests themselves. The only valid information in group (c) graph is the tentacles of the pests, and the body parts are completely occluded. In this scenario, the improved YOLOv12 model introduces the EMA attention mechanism, while the ECA attention mechanism is introduced in the C3k2-B module. These, together, direct the model to focus on the effective features, such as the unobstructed antennae and the color of the shells of the pests, and improve the model’s ability to extract and recognize the features of the pests under the occlusion situation. Based on the results from three sets of tests, the improved YOLOv12 model achieves higher confidence scores than the original model and YOLOv11 when dealing with pest occlusions. In single-object detection scenarios, it does not exhibit the duplicate detection issues present in the original model.

### 4.2. Limitations of the Model in This Paper

Although the model proposed in this study achieved better detection results, it also has these limitations. In Section 4.1, the model addresses the issue of repeated detection in scenarios such as low pest density. However, when confronted with complex samples such as overlapping pests (e.g., whiteflies) or morphologically diverse ones (e.g., leafminers), the model still tends to produce suboptimal detection boundaries. Visualization results are shown in Figure 17 (highlighted with yellow circles). In group (a), the model recognizes two whiteflies in close proximity to each other as a single. In group (b), the model exhibits suboptimal bounding box localization when identifying minute, densely clustered whiteflies and extensive leafminers’ tunnels lacking distinct individual delineation, resulting in overlapping instances. In group (c), the model may have duplicate detections with different detection frames overlapping each other when confronted with leafhopper tunnels with complex and diverse morphology. In this case, the phenomenon occurring in the (a) (c) group of graphs can have an impact on the counting of pest populations.

For high-density whitefly populations, the reasons for suboptimal detection results are as follows: First, the model’s preset anchor frame dimensions were designed based on statistical averages of five pest categories. K-means clustering generated three anchor frame groups, which failed to adequately cover the minute size and high-density distribution of whiteflies. This resulted in assignment collisions among adjacent instances within the same anchor frame, leading to insufficient positive samples for some instances. Meanwhile, the characteristic response area of whiteflies occupies only 1–3% of the image, and the coordinate update gradient during anchor box regression is weak, making it difficult to achieve precise segmentation of adjacent targets through bounding box fine-tuning. Additionally, there are some shortcomings in the adaptation of the receptive field and the minimum feature step size. The backbone network of YOLOv12 employs an improved C2f module, whose receptive field is better suited for medium-sized objects. For high-density aphids, the feature information of individual targets is compressed, typically corresponding to only one or two units in the finest-grained feature map. When the minimum step size is large, the contrast and separability of minute objects in high-level features diminish, making it difficult to distinguish subtle distances between adjacent individuals. Consequently, in high-density scenes, adjacent objects are more likely to be merged into a single prediction box. Finally, label assignment and post-processing strategies have limitations. The IoU-based allocation method tends to favor single instances, while fixed-threshold Greedy Non-Maximum Suppression (NMS) is prone to incorrectly merging adjacent detection boxes into a single box when small targets overlap heavily. This ultimately leads to issues such as merged adjacent instances, overlapping annotated boxes, and reduced localization accuracy.

For targets such as leafminers’ tunnels, which exhibit diverse morphologies and lack clear physical boundaries, the reasons for suboptimal detection performance are as follows: First, annotation technology has certain limitations. The leafminers’ tunnels are winding and irregular, lacking distinct individual boundaries. Each image displays different tunnel patterns, making it difficult to establish consistent annotation standards and hindering the model’s ability to learn stable target features. Meanwhile, rectangular annotation boxes struggle to conform to the slender and curved shapes of the tunnels. The optimal rectangle inevitably covers more background, increasing background response and consequently affecting localization and confidence. Second, feature modeling inadequately represents channel-like structures. The model adopts the shared feature architecture commonly used in the YOLO series for classification and regression. The backbone and neck primarily rely on generic texture and contour cues, lacking feature extraction for slender structures. Since the classification branch relies more on local texture cues (such as leaf damage), while the regression branch depends more on global geometry (such as the start/end points and curvature trends of tunnels), the two branches share features without being decoupled. This leads to feature interference on morphologically variable targets like leafminers’ tunnels, resulting in the occurrence of duplicate boxes.

In subsequent research, we will address the aforementioned technical limitations and model structure constraints by incorporating instance segmentation techniques and optimizing anchor box size design. Additionally, by expanding and enhancing the dataset of challenging samples in the aforementioned scenarios and annotating more precise bounding boxes, the model’s ability to recognize and segment high-density and morphologically diverse objects will be strengthened.

## 5. Summary

Cantaloupe is susceptible to a wide range of pests during growth, affecting agroecology and fruit yield. Therefore, this study realizes the detection of five types of cantaloupe leaf pests based on the improved YOLOv12 model, and the experiments show that the mAP50 of the improved YOLOv12 model reaches 85.06%, the P is 86.76%, the R is 79.94%, the mAP50-95 is 54.15%, and the F_1_ is 83.08%, which are improved from the original network, respectively, by 3.48%, 3.17%, 2.5%, 3.45% and 3.04%, and the number of parameters of the improved model is slightly reduced from 2.52 M to 2.43 M, and the amount of computation is only slightly increased. Therefore, the improvement strategy proposed in this study better balances the detection capability and lightweight performance of the model, and outperforms other YOLO series and RT-DETR models in terms of comprehensive performance. Not only that, but the improved model continues to perform well in detecting corn and rice pests, demonstrating strong generalization and applicability, thereby providing valuable reference for field pest control. However, there are still some optimizable problems in the improved model. Firstly, after the improvement of YOLOv12, the AP value of leafminers was increased from 58.4% to 62.8%, but there is still a big gap compared with the other four types of pests. Due to the special life habits of leafminers, their expression on the leaves of cantaloupe is in the form of zigzagging worm paths, irregular morphology, and different lengths and thicknesses. These features lead to a large amount of background information inevitably contained in the labeling box, and the background information will be different in the different growth periods of the leaves. The above factors cause the model to face greater difficulties in extracting target features, and it is prone to missed detection and repeated detection. Therefore, future work will firstly further investigate how to better extract the characteristic information of this category of pests, such as summarizing the characteristics of the pest dataset by introducing the knowledge of channel texture, length and distribution characteristics based on the experience in related fields, and secondly increasing the number of samples of the leafminers category, which contains different leaf backgrounds and different morphologies of the tunnels. Additionally, the improved model occasionally produces suboptimal detection boxes when encountering complex object detection scenarios. Future enhancements will involve constructing a dataset of challenging samples, annotating more precise bounding boxes, and incorporating instance segmentation techniques alongside optimized anchor box size designs to strengthen the model’s learning capabilities in complex environments. Finally, the improved YOLOv12 model needs to be further validated and optimized for detection on other crops to ensure its good generalization and applicability.

## Figures and Tables

**Figure 1 insects-16-01201-f001:**
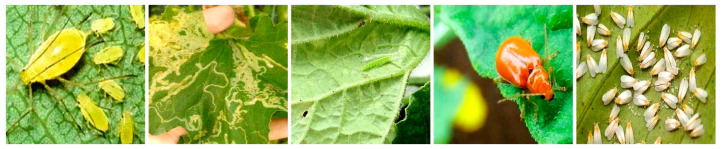
From left to right, Aphids, Leafminers (tunnels), Moths, Red-Melon Beetles, and Whiteflies.

**Figure 2 insects-16-01201-f002:**
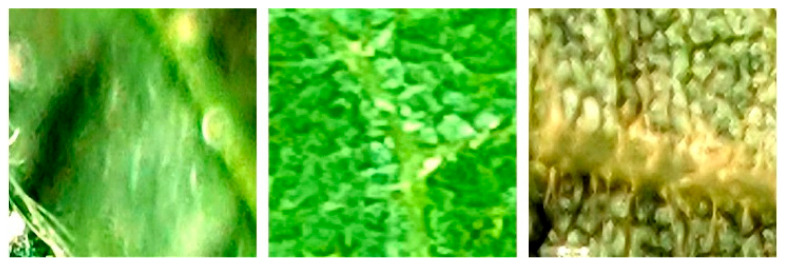
Some of the removed problematic images (from left to right, blurred, distorted, and unlabeled images).

**Figure 3 insects-16-01201-f003:**
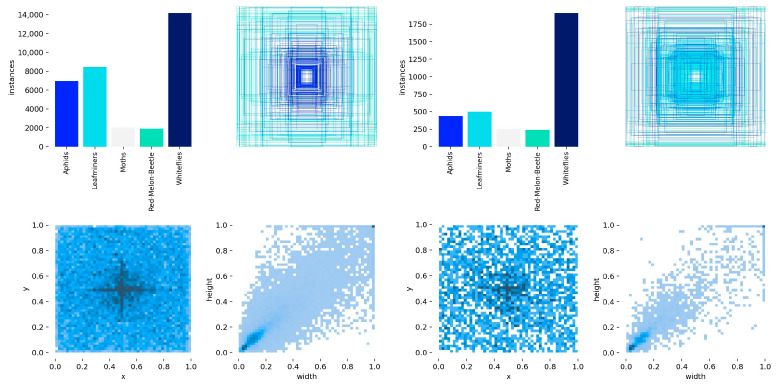
Training set and validation set label information.

**Figure 4 insects-16-01201-f004:**
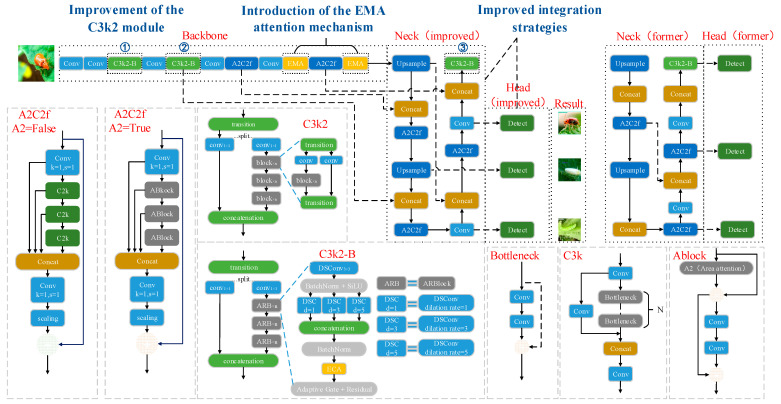
Diagram of the improved network structure of YOLOv12. Markers ①, ②, and ③ indicate the locations where the improved C3k2 module is applied. All arrows represent the flow of feature maps, whereas dashed arrows denote feature flows directed to non-adjacent modules.

**Figure 5 insects-16-01201-f005:**
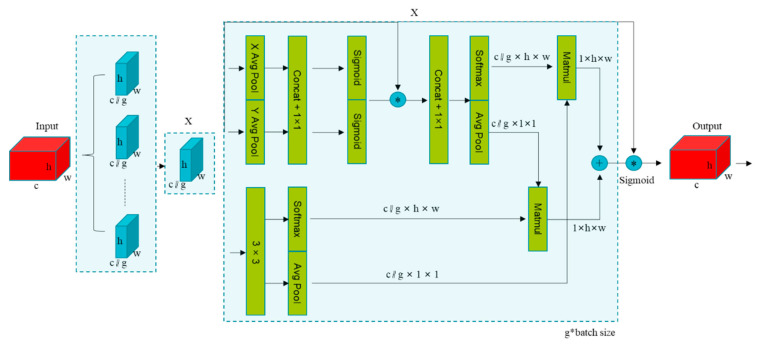
Structure diagram of the EMA attention mechanism. Arrows indicate feature flow direction; * denotes element-wise multiplication; + denotes element-wise addition.

**Figure 6 insects-16-01201-f006:**
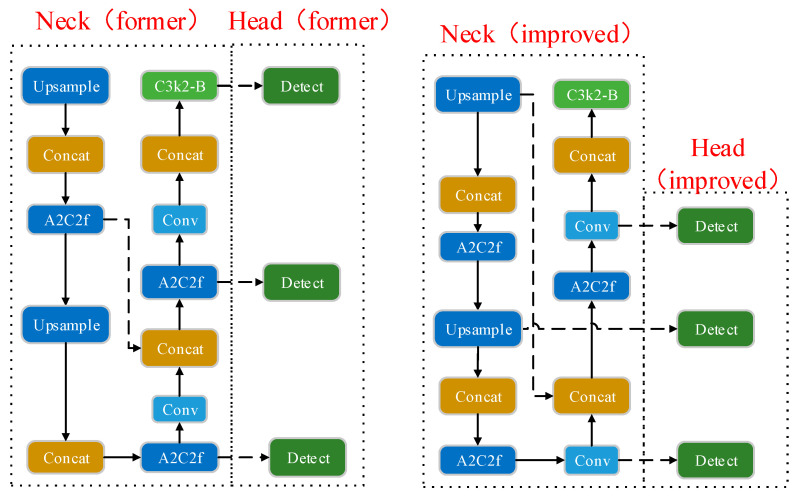
Neck and Detect structure before (**left**) and after (**right**) improvement. All arrows represent the flow of feature maps, whereas dashed arrows denote feature flows directed to non-adjacent modules.

**Figure 7 insects-16-01201-f007:**
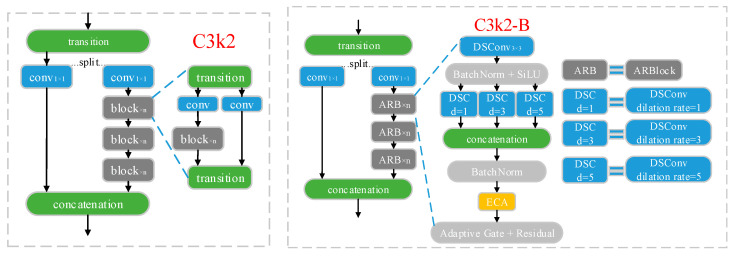
Structure of C3k2 module before (**left**) and after (**right**) improvement. The term “split” denotes the operation that partitions the input feature map along the channel dimension.

**Figure 8 insects-16-01201-f008:**
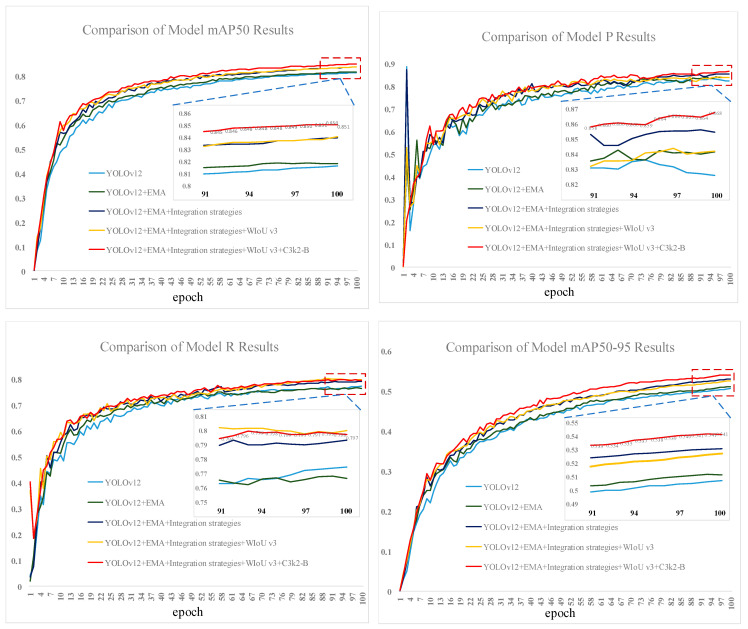
Comparison of training metrics results for models 1, 2, 6, 7, and 8 in the ablation study.

**Figure 9 insects-16-01201-f009:**
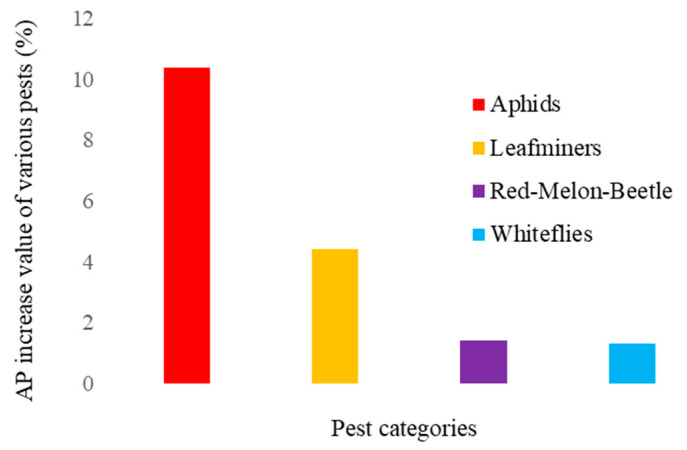
AP increase values for different types of pests.

**Figure 10 insects-16-01201-f010:**
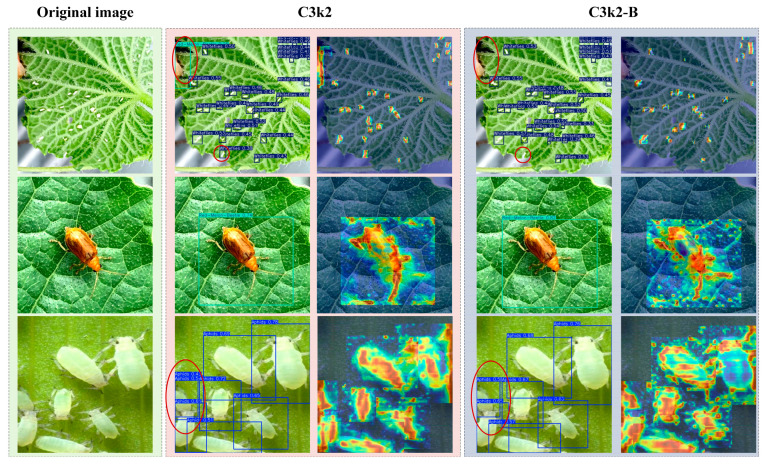
Comparison of detection results and heat maps between the C3k2 and C3k2-B models.

**Figure 11 insects-16-01201-f011:**
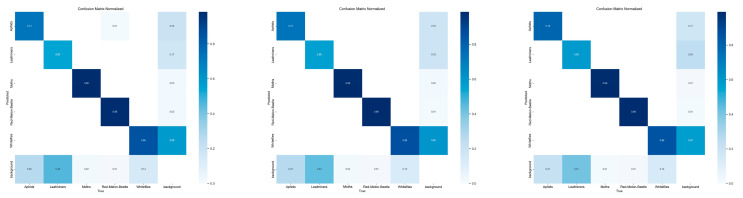
From left to right: confusion matrices for YOLOv12, YOLOv11, and the improved YOLOv12.

**Figure 12 insects-16-01201-f012:**
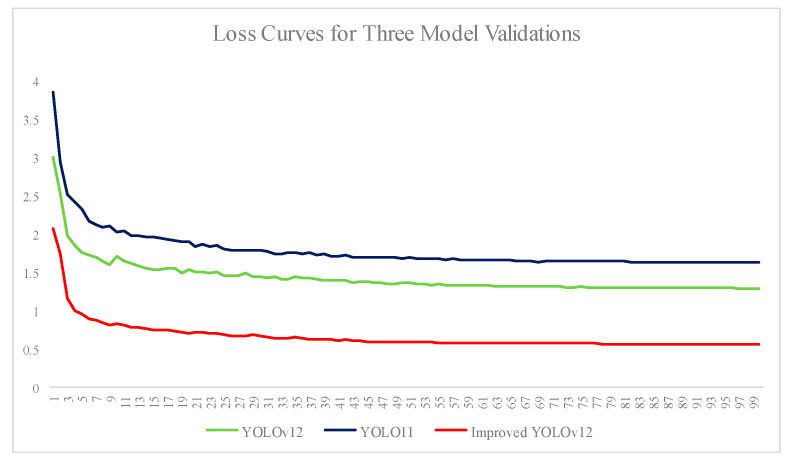
Validation loss curves for YOLOv12, YOLOv11, and the improved YOLOv12.

**Figure 13 insects-16-01201-f013:**
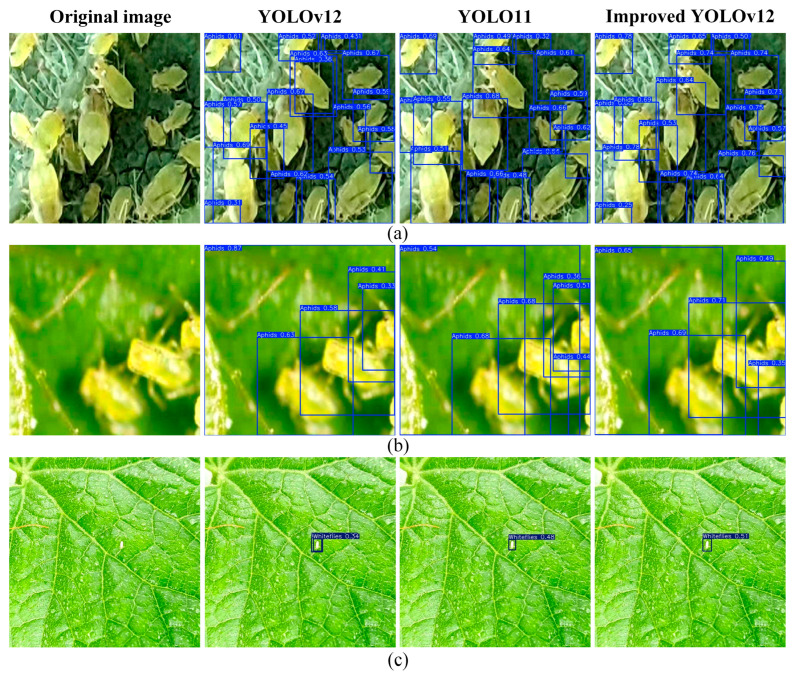
Comparison of detection results for pests of different sizes using three models (YOLOv12, YOLO, and Improved YOLOv12). (**a**,**b**) aphids are densely distributed, vary in size, and are partially obscured; (**c**) tiny whiteflies.

**Figure 14 insects-16-01201-f014:**
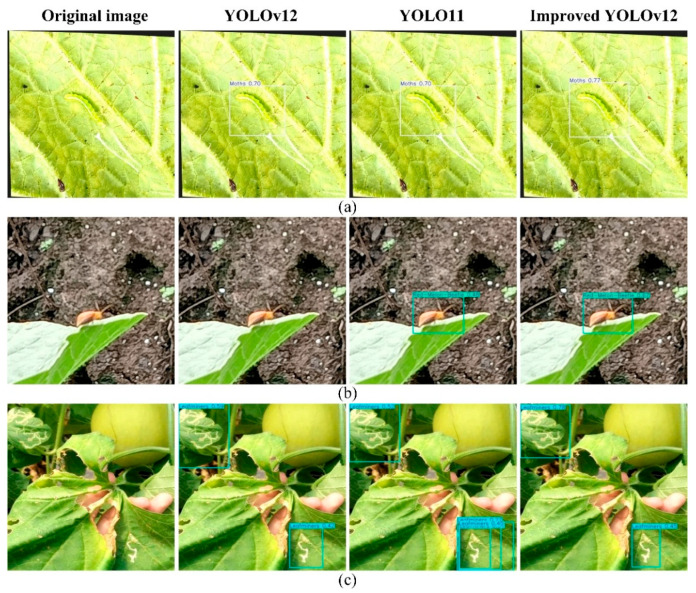
Comparison of detection results for different background pests using three models (YOLOv12, YOLO, and Improved YOLOv12). (**a**) the moth’s color blends with its background; (**b**) the image contains background information such as leaves and the ground; (**c**) The image contains complex backgrounds such as leaves, roots, stems, and fruits.

**Figure 15 insects-16-01201-f015:**
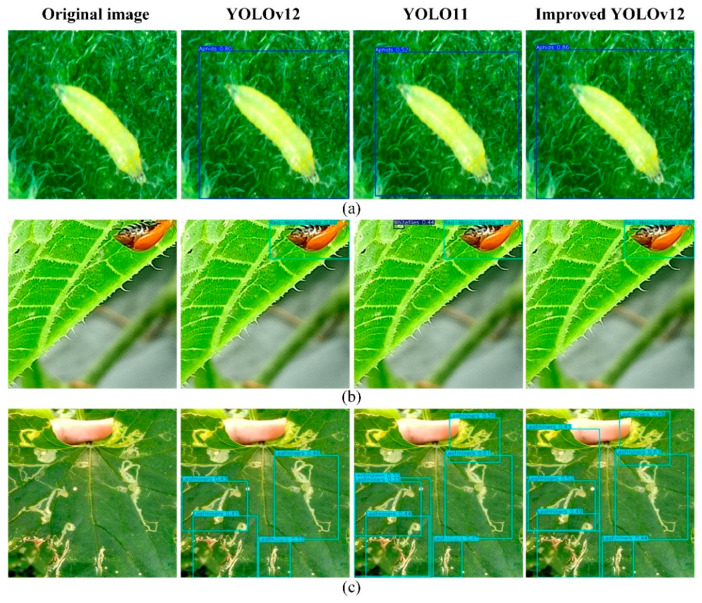
Comparison of detection results for different target shapes among three models (YOLOv12, YOLO, and Improved YOLOv12). (**a**) aphids with unusual appearances; (**b**) the red-melon beetle with its unique posture; (**c**) the diverse tunnels of leafminers.

**Figure 16 insects-16-01201-f016:**
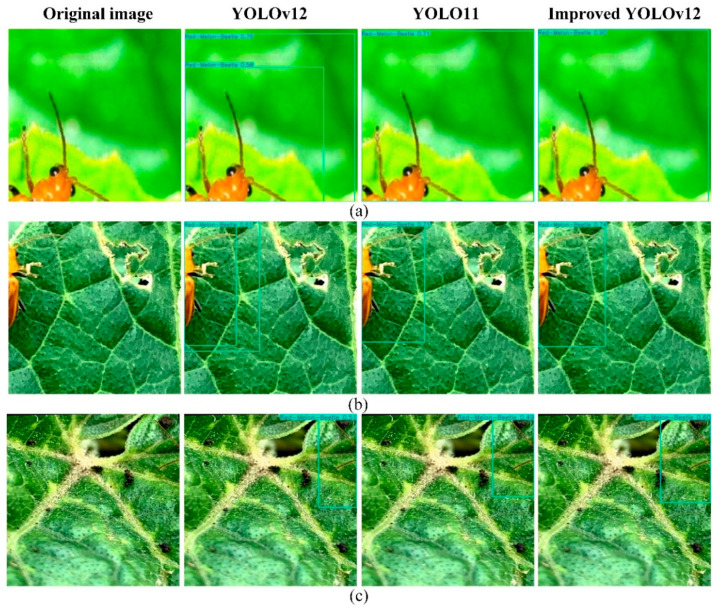
Comparison of detection results for pests with different levels of occlusion using three models (YOLOv12, YOLO, and Improved YOLOv12). (**a**) moderately obscured red-melon beetle; (**b**) Highly obscured red-melon beetle; (**c**) the nearly completely obscured red-melon beetle.

**Figure 17 insects-16-01201-f017:**
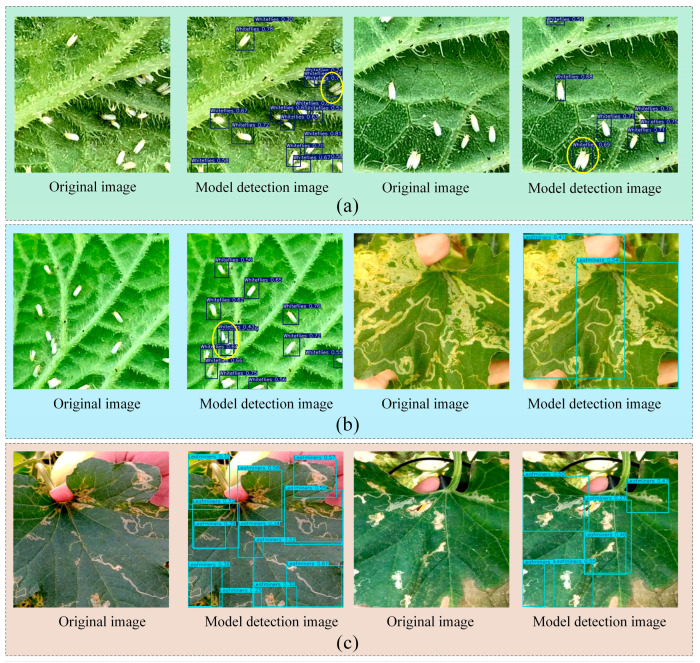
Visualization of the limitations of the improved YOLOv12 model in this study: (**a**) identification of immediately adjacent pests as single pests (marked with yellow circles in the image); (**b**) overlap of detection boxes with too large a range; (**c**) duplication of labeled boxes.

**Table 1 insects-16-01201-t001:** Labels for various pests and data distribution.

Pest Categories	Labels	Number of Training Set Images	Number of Validation Set Images	Number of Test Set Images
Aphids	0	1904	119	119
Leafminers	1	2200	137	139
Moths	2	2005	250	252
Red-Melon Beetles	3	1873	234	235
Whiteflies	4	1985	248	249
Total	-	9967	988	994

**Table 2 insects-16-01201-t002:** Parameter configuration table.

Parameter Name	Parameter Value
epoch	100
image size	640 × 640
batch size	8
momentum	0.937
lr	0.01
decay	0.0005
optimizer	SGD

**Table 3 insects-16-01201-t003:** Ablation experiment.

Models	EMA	Integration Strategies	WIoU v3	C3k2-B	mAP50/%	P/%	R/%	mAP50-95/%	Parameters/M	Computational Volume/GFLOPs
1	×	×	×	×	81.58	83.59	77.44	50.70	2.52	6.0
2	√	×	×	×	81.86	84.23	76.77	51.15	2.54	6.1
3	×	√	×	×	82.78	83.94	78.96	52.31	2.51	6.7
4	×	×	√	×	82.06	82.75	77.58	51.56	2.52	6.0
5	×	×	×	√	82.24	83.88	78.49	51.73	**2.42**	**5.9**
6	√	√	×	×	83.95	85.58	79.31	53.07	2.53	6.9
7	√	√	√	×	84.04	84.34	**80.01**	52.70	2.53	6.9
8	√	√	√	√	**85.06**	**86.76**	79.94	**54.15**	2.43	6.7

Note: √ indicates that this improvement strategy was introduced, × indicates that it was not introduced, and all results in the table are retained to two decimal places. Bold indicates the best result.

**Table 4 insects-16-01201-t004:** Detection results of the improved strategy for five pest categories.

Pest Categories	YOLOv12	+EMA	+Integration Strategies	+WIoU v3	+C3k2-B	+All Improvement Strategies
Aphids	69.0	71.3	71.8	70.2	71.2	**79.4**
Leafminers	58.4	57.3	60.2	59.8	58.7	**62.8**
Moths	98.9	98.4	98.4	98.3	98.6	**98.9**
Red-Melon Beetles	97.5	97.4	97.8	98.4	98.4	**98.9**
Whiteflies	84.0	84.6	85.3	83.4	84.3	**85.3**

Note: All results in the table are retained in one decimal place and are expressed in %. Bold indicates the best result.

**Table 5 insects-16-01201-t005:** Comparison experiment with other models.

Models	mAP50/%	P/%	R/%	mAP50-95/%	F_1_/%	Parameters/M	Computational Volume/GFLOPs
YOLOv5n-7.0	80.60	80.56	80.29	43.22	80.19	**1.77**	**4.2**
YOLOv5s-7.0	83.62	83.56	**82.69**	47.7	82.66	7.03	16.0
YOLOv6n	82.64	84.56	79.38	51.72	81.89	4.24	11.9
YOLOv8n	83.16	83.79	79.84	52.18	81.46	3.01	8.2
YOLOv10n	82.01	85.11	78.09	51.55	78.71	2.71	8.4
YOLO11n	83.11	84.23	78.94	52.78	81.32	2.59	6.4
YOLOv12n	81.58	83.59	77.44	50.70	80.04	2.52	6.0
RT-DETR-l	82.55	84.68	80.42	50.88	82.06	32.82	108.0
Improved YOLOv12 (Ours)	**85.06**	**86.76**	79.94	**54.15**	**83.08**	2.43	6.7

Note: All results in the table are retained to two decimal places. Bold indicates the best result.

**Table 6 insects-16-01201-t006:** Comparative experiment of different attention mechanisms.

Replacement of Attention Mechanisms	mAP50/%	P/%	R/%	mAP50-95/%	Parameters/M	Computational Volume/GFLOPs
former EMA	**85.06**	**86.76**	79.94	**54.15**	2.43	6.7
CA	85.00	86.62	80.24	53.89	2.42	6.6
SimAM	84.16	85.75	80.07	53.28	**2.41**	**6.6**
ECA	83.95	85.06	79.72	53.16	**2.41**	**6.6**
CBAM	84.55	85.49	79.82	53.84	2.54	6.7
SE	84.22	84.55	**80.27**	53.37	2.42	6.6
SA [35]	84.52	86.30	79.62	53.22	**2.41**	**6.6**

Note: All results in the table are retained to two decimal places. Bold indicates the best result.

**Table 7 insects-16-01201-t007:** Comparison of basic information for three datasets.

Dataset Name	Scene Characteristics	Number of Images in the Training Set	Image Content Details
Melon Cantaloupe Pest	Focusing on leaf scenes at different growth stages of honeydew melons. Moth pests are occasionally observed on the stems.	9967 sheets	Pest instances exhibit relatively high density, with an average of 3.5 targets per image. Images of aphids, whiteflies, and leafminers contain anywhere from several to over a dozen targets. The dataset covers varying light intensities (bright and dim lighting) and rainy conditions, and includes pests of the same species but differing in morphology and size.
Pest-dataset	Covering corn kernels, leaves, stalks, soil of varying colors, and palms, among other scenarios, to simulate field and manual observation environments.	2730 sheets	Most images contain only one or two pests, with an average of 1.5 specimens per image; some images are composed of multiple specimens of the same pest species.
Rice Pests	Covering rice leaves, stems, grains, and palm scenes, some of which are artificially constructed photography settings.	3922 sheets	Most images contain only one or two pests, include environments with low light levels, and feature both close-up and distant views of the same pest species.

**Table 8 insects-16-01201-t008:** Generalization experiment 1.

Models	mAP50/%	P/%	R/%	mAP50-95/%	Parameters/M	Computational Volume/GFLOPs
Improved YOLOv12	**94.08**	**94.63**	**89.15**	**66.99**	**2.43**	6.8
YOLOv12	88.64	88.49	81.03	63.49	2.52	**6.0**

Note: All results in the table are retained to two decimal places. Bold indicates the best result.

**Table 9 insects-16-01201-t009:** Generalization experiment 2.

Models	mAP50/%	P/%	R/%	mAP50-95/%	Parameters/M	Computational Volume/GFLOPs
Improved YOLOv12	**90.91**	**91.17**	**85.60**	**60.17**	**2.43**	6.8
YOLOv12	86.09	86.83	79.98	54.67	2.52	**6.0**

Note: All results in the table are retained to two decimal places. Bold indicates the best result.

## Data Availability

[Melon Cantaloupe Pest] author: Trisha Mae; title: Melon Cantaloupe Pest Dataset; publisher: Roboflow; howpublished: https://universe.roboflow.com/trisha-mae/melon-cantaloupe-pest (accessed on 30 March 2025); journal: Roboflow Universe. [Generalization experiment 1] author: Varshine; title: Pest-dataset Dataset; publisher: Roboflow; howpublished: https://universe.roboflow.com/varshine/pest-dataset-l3ph3 (accessed on 16 June 2025); journal: Roboflow Universe. [Generalization experiment 2] author: Laktharu; title: Rice Pests Dataset; publisher: Roboflow; howpublished: https://universe.roboflow.com/laktharu/rice-pests-ztbeq (accessed on 1 June 2025); journal: Roboflow Universe.

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
