# Peer review of "Multi-Strategy Improved Cantaloupe Pest Detection Algorithm"

_insects, 2025, doi:10.3390/insects16121201_

Round 1

Reviewer 1 Report

Comments and Suggestions for Authors
  1. The authors should review their document, as there are several grammatical and spelling errors, including word repetition and issues when constructing sentences.
  2. The authors had a big data set and is adequate for the implementation of Yolo
  3. It would be awesome if they could consider in this work adding more performance metrics like accuracy, F1, sensitivity and specificity in addition to the metrics mentioned above, or at least the confusion matrix(es).
  4. The graphs shown in Figure 7 are very important for the study, for this reason, I strongly recommend that they could be in a better resolution. At this point it is almost impossible to understand the graph.

Author Response

1. The authors should review their document, as there are several grammatical and spelling errors, including word repetition and issues when constructing sentences.

Thank you for your suggestion. We reviewed our article section by section, correcting grammatical and spelling errors, as well as addressing word repetition and sentence structure issues. We have also marked the modified sections.

2. The authors had a big data set and is adequate for the implementation of Yolo

Thank you to the reviewers for your approval.

3. It would be awesome if they could consider in this work adding more performance metrics like accuracy, F1, sensitivity and specificity in addition to the metrics mentioned above, or at least the confusion matrix(es).

We appreciate the reviewer's suggestions and are pleased to incorporate them. In the section comparing different models (Table 5, Page 14, Line 483), we have added the F1 score metric and included all relevant content. However, without considering accuracy, we believe that in the field of object detection (specifically melon pest detection, which is our research focus), mAP (mean average precision) serves as a superior alternative to accuracy metrics. It is particularly well-suited for scenarios involving multiple categories and precise localization, and carries greater authority;Sensitivity and recall (R) are essentially the same, and we have also provided them in our experiments; Specificity reflects a model's ability to distinguish pests from background elements. In practical testing, we found that nearly all models could detect the target, with only rare instances of misclassifying background as pests. Since model specificity showed virtually no variation, we did not include this metric. Additionally, we provide the confusion matrices for the improved YOLOv12 model and its suboptimal counterpart YOLOv11 (Figure 11,Page 15, Lines 488-489 ), along with the validation loss curves (Figure 12, Page15, Lines 495) and accompanying descriptions (Figure 12 shows the validation loss curves for YOLOv12, the suboptimal model YOLOv11, and the improved YOLOv12. As shown in the figure, the loss of all three mod-els gradually decreases with increasing training iterations, eventually converging to a sta-ble value. Notably, the improved YOLOv12 model converges to a lower value. Page 15, Lines 490-493), to better demonstrate the models' performance.In Section 3.2, Model performance evaluation metrics, the definition of F1 has been added: “F1 is the weighted harmonic mean of Precision and Recall.”(Page 10, Lines 358-360)

4. The graphs shown in Figure 7 are very important for the study, for this reason, I strongly recommend that they could be in a better resolution. At this point it is almost impossible to understand the graph.

Thank you for your suggestion. We have redrawn Figure 7 (now Figure 8), enhanced its resolution, and enlarged the metric curves for the 10 rounds following model training. We have also added specific numerical values for all four metrics of our improved model: mAP50, Precision, Recall, and mAP50-95.(Page 12, Lines 425)

Reviewer 2 Report

Comments and Suggestions for Authors
  1. The dataset, validation set, and test set exhibit class imbalance. It is recommended to provide a detailed distribution of pest samples and whether any targeted data augmentation strategies were attempted.
  2. The data preprocessing section mentions augmentations such as horizontal flipping for aphids and leaf miners. It is suggested to provide a detailed augmentation plan and the resulting changes in data distribution to demonstrate the completeness of the dataset optimization.
  3. The generalization experiments use pest datasets from corn and rice, but the annotation quality, scene diversity, and differences compared to the melon pest dataset are not clearly stated. It is recommended to include a comparison of basic dataset information to enhance the credibility of the generalization ability validation.
  4. In the model limitations section, it is suggested to delve deeper into the technical factors causing the issue, combining them with the model structure for a more comprehensive discussion.
  5. In the comparison of different attention mechanisms, although the EMA mechanism performs better in core metrics, mechanisms like CA and SE show slight improvements in recall. It is suggested to provide an analysis of the characteristics of different attention mechanisms and their suitability for pest detection tasks, explaining the rationale behind choosing EMA more comprehensively.
  6. Some figures (such as Figure 7, Figures 10-14) lack detailed legends. It is recommended to add textual explanations of the key information in these charts.
  7. The reference formatting is inconsistent. Some Chinese references lack DOIs or publication information, and the formatting of English references is not unified. The reference format should be standardized according to journal requirements.
  8. It is recommended to specify optimization plans for issues such as low detection accuracy for leaf beetles and inaccurate bounding boxes in high-density scenes in future research. Considerations like introducing instance segmentation technology or optimizing anchor box size design could be addressed.
  9. The model's deployment testing in real-field environments can be added, including hardware compatibility, real-time detection speed (FPS) on embedded devices, and other aspects to further validate the model's engineering application value.

Author Response

1. The dataset, validation set, and test set exhibit class imbalance. It is recommended to provide a detailed distribution of pest samples and whether any targeted data augmentation strategies were attempted.

Thank you for the reviewers' suggestions. During the data preprocessing stage, we first excluded samples with severe blurring, distortion, or missing annotations. To prevent data leakage, we then divided the dataset into training, validation, and testing sets at an 8:1:1 ratio per category before applying data augmentation. The augmentation method is directional augmentation. To mitigate imbalance and ensure evaluation fairness, we only perform data augmentation on the two minority classes within the training set: the number of aphid images increased from 952 to 1904, while leafminer images rose from 1100 to 2200 (with 50% horizontally flipped and 50% vertically flipped). The number of images in both the validation set and test set remains unchanged, with no augmentation applied, to ensure the objectivity of the model's generalization capability assessment. We have provided the number of each pest category in Table 1, and have added Figure 3 to illustrate the detailed distribution of each pest sample.

2. The data preprocessing section mentions augmentations such as horizontal flipping for aphids and leaf miners. It is suggested to provide a detailed augmentation plan and the resulting changes in data distribution to demonstrate the completeness of the dataset optimization.

Thank you for the reviewers' suggestions.Thank you for the reviewers' suggestions. We provided a detailed enhancement plan: 50% of the training set images for aphids and leafminers were horizontally flipped, while the remaining 50% were vertically flipped. The number of images in the validation and test sets remained unchanged. In Section 2.1, we supplemented the number of images before and after expansion: “To mitigate classifier bias and enhance dataset diversity, data augmentation techniques involving horizontal flipping and vertical flipping (each applied with a 50% probability) were employed to augment the training images for aphids and leafminers, which had relatively smaller datasets. The aphids training dataset expanded from 952 to 1,904 images, while the leafminers dataset grew from 1,100 to 2,200 images.”(Lines 148-152) The original dataset already incorporates numerous image enhancement techniques, including preprocessing with automatic orientation and size normalization alongside contrast optimization, as well as random image enhancements such as 90° random rotation, 0-20% random cropping, ±15% random brightness variation, and 0-1.4 pixel random blurring. Additionally, the Melon Cantaloupe Pest dataset contains a substantial volume of data with rich pest characteristics. It includes leaf pests at various growth stages and features sufficiently challenging samples. These encompass scenarios such as partially obscured pests, rainy-day environments, diverse poses, varying pest sizes, dense pest distributions, and both close-up and distant shots—all of which provide ample material for model training. Therefore, we only performed simple augmentation on the training sets for the aphids and leafminers mentioned above to prevent their numbers from diverging too greatly from the other three categories.

3. The generalization experiments use pest datasets from corn and rice, but the annotation quality, scene diversity, and differences compared to the melon pest dataset are not clearly stated. It is recommended to include a comparison of basic dataset information to enhance the credibility of the generalization ability validation.

We appreciate the reviewer's suggestions and have clarified the annotation quality of the dataset in Section 3.7: “We screened the images and annotations. Both datasets feature high annotation quality, appropriately sized bounding boxes, and no instances of missing annotations or excessively blurred images.”(Lines 553-555) Furthermore, we compared the basic information of the three datasets—Melon Cantaloupe Pest, Pest-dataset, and Rice Pests—across three dimensions: scene characteristics, number of training images, and image content detail. This comparison is summarized in Table 7 (Line 568).

4. In the model limitations section, it is suggested to delve deeper into the technical factors causing the issue, combining them with the model structure for a more comprehensive discussion.

We appreciate the reviewers' suggestions and have conducted an in-depth investigation into the causes of the model's limitations, covering both technical factors and model structure. In Section 4.2, our analysis is as follows:“For high-density whitefly populations, the reasons for suboptimal detection results are as follows: First, the model's preset anchor frame dimensions were designed based on statistical averages of five pest categories. K-means clustering generated three anchor frame groups, which failed to adequately cover the minute size and high density distribu-tion of whiteflies. This resulted in assignment collisions among adjacent instances within the same anchor frame, leading to insufficient positive samples for some instances. Meanwhile, the characteristic response area of whiteflies occupies only 1%-3% of the im-age, and the coordinate update gradient during anchor box regression is weak, making it difficult to achieve precise segmentation of adjacent targets through bounding box fi-ne-tuning. Additionally, there are some shortcomings in the adaptation of the receptive field and the minimum feature step size. The backbone network of YOLOv12 employs an improved C2f module, whose receptive field is better suited for medium-sized objects. For high-density aphids, the feature information of individual targets is compressed, typically corresponding to only one or two units in the finest-grained feature map. When the mini-mum step size is large, the contrast and separability of minute objects in high-level fea-tures diminish, making it difficult to distinguish subtle distances between adjacent indi-viduals. Consequently, in high-density scenes, adjacent objects are more likely to be merged into a single prediction box. Finally, label assignment and post-processing strate-gies have limitations. The IoU-based allocation method tends to favor single instances, while fixed-threshold Greedy Non-Maximum Suppression (NMS) is prone to incorrectly merging adjacent detection boxes into a single box when small targets overlap heavily. This ultimately leads to issues such as merged adjacent instances, overlapping annotated boxes, and reduced localization accuracy.

For targets such as leafminers’ tunnels, which exhibit diverse morphologies and lack clear physical boundaries, the reasons for suboptimal detection performance are as fol-lows: First, annotation technology has certain limitations. The leafminers’ tunnels are winding and irregular, lacking distinct individual boundaries. Each image displays dif-ferent tunnel patterns, making it difficult to establish consistent annotation standards and hindering the model's ability to learn stable target features. Meanwhile, rectangular anno-tation boxes struggle to conform to the slender and curved shapes of the tunnels. The op-timal rectangle inevitably covers more background, increasing background response and consequently affecting localization and confidence. Second, feature modeling inadequate-ly represents channel-like structures. The model adopts the shared feature architecture commonly used in the YOLO series for classification and regression. The backbone and neck primarily rely on generic texture and contour cues, lacking feature extraction for slender structures. Since the classification branch relies more on local texture cues (such as leaf damage), while the regression branch depends more on global geometry (such as the start/end points and curvature trends of tunnels), the two branches share features without being decoupled. This leads to feature interference on morphologically variable targets like leafminers’ tunnels, resulting in the occurrence of duplicate boxes.”(Lines 681-720)

5. In the comparison of different attention mechanisms, although the EMA mechanism performs better in core metrics, mechanisms like CA and SE show slight improvements in recall. It is suggested to provide an analysis of the characteristics of different attention mechanisms and their suitability for pest detection tasks, explaining the rationale behind choosing EMA more comprehensively.

We appreciate the reviewer's suggestions. In Section 3.6, “Comparative experiments of different attention mechanisms,” we reviewed the literature on CA, SE, SimAM, and EMA attention mechanisms, introduced their principles and advantages in the melon pest detection task, and analyzed why CA, SE, and SimAM achieved higher R scores than EMA while performing lower on the other three metrics. We also proposed how to compensate for EMA's shortcomings in detection scenarios that prioritize R. We have added the following statement to Section 3.6:“Additionally, it is worth noting that the R improved slightly after introducing the CA, SimAM, and SE attention mechanisms. The SE (Squeeze and Excitation) attention mechanism obtains channel descriptions through global average pooling and uses two fully connected layers to generate channel recalibration weights, significantly enhancing inter-channel dependencies. SE enhances the overall response of the channel, amplifying weak targets such as whiteflies. However, due to the lack of fine-grained spatial selection, background textures may also be amplified simultaneously. Compared to EMA, this approach tends to yield a slight increase in R but a slight decrease in P and mAP. The CA (Coordinate Attention) mechanism decomposes the 2D global pooling operator into two 1D directional encodings along the height and width axes. This preserves precise positional information while establishing long-range dependencies, ultimately generating direction-aware and position-sensitive channel attention maps. CA embeds directional position information within the channel attention mechanism, making it more sensitive to targets with directional structures (such as leafminers’ tunnels). This enhances its ability to extract trajectory features from weak targets, thereby improving R. However, its spatial gating remains primarily channel-based, and its suppression of complex backgrounds is less effective than multi-scale pixel-level feature modeling. SimAM (Simple Attention Module) calculates 3D attention weights for each spatial position-channel dimension based on the principle of minimizing the neural energy function, without requiring additional parameters. SimAM employs energy-based pointwise weighting, which globally enhances salient responses (such as the red coloration of the red melon beetles), typically resulting in a slight improvement in R. However, due to the lack of explicit multi-scale contextual modeling and cross-spatial interactions, it is challenging to effectively distinguish the boundaries and features of adjacent targets in scenarios where targets such as whiteflies and aphids are clustered. Consequently, its contribution to mAP is relatively limited. EMA is a multi-scale contextual attention mechanism that balances local details with global semantics. It suppresses background noise and focuses on small target features in scenarios with small objects, large scale variations, and complex backgrounds. This enables the model to more accurately identify pest species and locations, making it better suited for the melon pest detection task. Consequently, it achieves the best performance across three metrics: mAP50, P, and mAP50-95. The R values for CA, SE, and SimAM attention mechanisms are slightly higher than those for EMA, as they enable the model to amplify weak target signals. However, all three suffer from insufficient spatial or multi-scale selectivity, meaning their relatively high R values also come with a higher rate of misclassification. If the deployment scenario prioritizes recall (e.g., early warning systems), the gap in R can be addressed by methods such as category-specific thresholds while maintaining the EMA.”(Lines 512-547)

6. Some figures (such as Figure 7, Figures 10-14) lack detailed legends. It is recommended to add textual explanations of the key information in these charts.

We appreciate the reviewers' suggestions. We have redrawn Figure 7 (now Figure 8) and renamed it: “Comparison of training metrics results for models 1, 2, 6, 7, and 8 in the ablation study.”(Line 425) We have also supplemented the legends for Figures 10–14 (now Figures 13–17).(Lines 594-595, 611-612, 630-631, 664-665, 728-730)

7. The reference formatting is inconsistent. Some Chinese references lack DOIs or publication information, and the formatting of English references is not unified. The reference format should be standardized according to journal requirements.

We appreciate the reviewers' suggestions. We have revised the references in accordance with your journal's requirements and the published articles in your journal. We have also modified the corresponding content in the introduction.

8. It is recommended to specify optimization plans for issues such as low detection accuracy for leaf beetles and inaccurate bounding boxes in high-density scenes in future research. Considerations like introducing instance segmentation technology or optimizing anchor box size design could be addressed.

We sincerely appreciate the reviewer's suggestions and are honored to incorporate them into our future research. We will explore incorporating instance segmentation techniques or optimizing anchor box size design to enhance the model. Additionally, We have also incorporated your suggestions into Section 4.2 on model limitations: "In subsequent research, we will address the aforementioned technical limitations and model structure constraints by incorporating instance segmentation techniques and optimizing anchor box size design. Additionally, by expanding and enhancing the dataset of challenging samples in the aforementioned scenarios and annotating more precise bounding boxes, the model's ability to recognize and segment high-density and morphologically diverse objects will be strengthened.”(Lines 721-726)

9. The model's deployment testing in real-field environments can be added, including hardware compatibility, real-time detection speed (FPS) on embedded devices, and other aspects to further validate the model's engineering application value.

We appreciate the reviewers' suggestions. After experimental testing, our improved model achieved an FPS value of 66.67 on the test set. In subsequent research, we will also expand model deployment testing in real-world environments. Currently, we are testing the LuckFox Pico Plus device, which features a single-core ARM Cortex-A73 64-bit processor integrated with NEON and a floating-point unit (FPU). Its embedded neural network processing unit (NPU) supports mixed-precision operations (int4, int8, int16) with a peak computational capability of 0.5 TOPS. However, due to the school's current work schedule, we regret that we are unable to complete the deployment in the short term. In subsequent phases of the project, we will conduct further field experiments and test our model in real-world scenarios. We appreciate your understanding.

Reviewer 3 Report

Comments and Suggestions for Authors
  1. The ablation study (Table 3) fails to clearly demonstrate the synergistic effect of all components. A clear contradiction: Model 6 (EMA + Integration) achieves 83.95% mAP50. When the C3k2-B module is added (Model 7), the performance drops to 83.64%. This indicates C3k2-B has a negative effect in this combination. The authors' attempt to explain this as "temporary incomplete adaptation" is a weak and unconvincing argument. Despite this, the authors later claim C3k2-B is an "indispensable part" of the strategy. This contradictory analysis suggests the authors lack a deep understanding of the complex interactions between their proposed modules. The combination of modules appears ad-hoc rather than theoretically grounded.
  2. In Section 4.1, the authors repeatedly claim their model improves the "duplicate detection problem". However, in Section 4.2, "Limitations of the model," the authors explicitly state the model suffers from "duplicate detections" and show an example in Figure 14c. This direct contradiction is unacceptable.

Author Response

1. The ablation study (Table 3) fails to clearly demonstrate the synergistic effect of all components. A clear contradiction: Model 6 (EMA + Integration) achieves 83.95% mAP50. When the C3k2-B module is added (Model 7), the performance drops to 83.64%. This indicates C3k2-B has a negative effect in this combination. The authors' attempt to explain this as "temporary incomplete adaptation" is a weak and unconvincing argument. Despite this, the authors later claim C3k2-B is an "indispensable part" of the strategy. This contradictory analysis suggests the authors lack a deep understanding of the complex interactions between their proposed modules. The combination of modules appears ad-hoc rather than theoretically grounded.

We appreciate the reviewers for pointing out our issues and acknowledge that Table 3 is not sufficiently persuasive. However, we wish to clarify that our actual sequence of module integration is as follows: (1) Introduction of the EMA attention mechanism; (2) Introduction of the fusion strategy; (3) Introduction of the WIoU v3 loss function; (4) Introduction of the C3k2-B module. We also mentioned in our paper the reasons for introducing the aforementioned modules. Following this sequence of ablation experiments, the core metric mAP50 showed varying degrees of improvement with each additional module. However, when drawing the network architecture in Figure 3, we labeled the C3k2-B module as (3) for convenience, using sequential numbers (1), (2), (3) to illustrate our improvements within the network structure (since the loss function modification could not be depicted in the network diagram). Consequently, we also present the experimental results for Model 7 in Table 3. This approach, however, does indeed result in the actual model design not being accurately represented. Finally, we modified the sequence of improvements in the ablation experiments to align with the actual protocol and recreated Table 3(Page 11,Line 418), Table 4(Page 13, Line 437) and Figure 7 (now Figure 8, Page 12, Line 425). Additionally, the order of introduction has been adjusted according to the implementation sequence of the improvement measures (2.2.3(Page 7, Line 247) and 2.2.4(Page 8, Line 292)).

2. In Section 4.1, the authors repeatedly claim their model improves the "duplicate detection problem". However, in Section 4.2, "Limitations of the model," the authors explicitly state the model suffers from "duplicate detections" and show an example in Figure 14c. This direct contradiction is unacceptable.

We acknowledge the reviewers' points and recognize that our wording was inadequate. What we wish to convey is that our model has improved upon the original model's issue of duplicate detection to a certain extent, though it has not been completely resolved. We have reformulated some of the conclusions from Section 4.1. This includes those in Section 4.1.1:“Compared to the original model and YOLOv11, the improved YOLOv12 model incorporates an EMA attention mechanism, enhancing its detection capability for small objects. Consequently, it achieves higher confidence scores, and the improved model does not exhibit the duplicate detection issues present in the original model.”(Page 19, Lines 604-608) Section 4.1.2:“In group (c), which contains complex backgrounds including leaves, roots, stems, and fruits, the number of detection targets is relatively low. In this scenario, the improved YOLOv12 model does not exhibit the duplicate detection issues seen in YOLOv11 and demonstrated higher confidence levels.”(Page 20, Lines 624-627) Section 4.1.4:“Based on the results from three sets of tests, the improved YOLOv12 model achieves higher confidence scores than the original model and YOLOv11 when dealing with pest occlusions. In single-object detection scenarios, it does not exhibit the duplicate detection issues present in the original model.”(Page 21, Lines 659-662) Furthermore, in Section 4.2, we specifically highlighted the scenarios where the model effectively mitigates duplicate detection issues and those where the problem persists:“In Section 4.1, the model improves the issue of repeated detection in scenarios such as low pest density. However, when confronted with complex samples such as overlapping pests (e.g., whiteflies) or morphologically diverse ones (e.g., leafminers), the model still tends to produce suboptimal detection boundaries. Visualization results are shown in Figure 17 (highlighted with yellow circles).”(Page 21, Lines 668-672) Finally, following suggestions from other reviewers, we analyzed the causes of the model's limitations.

Reviewer 4 Report

Comments and Suggestions for Authors

The paper proposes an enhanced version of the original YOLOv12 architecture specifically designed for pests detection. Here are some considerations.

1. The authors should provide the innovation as a bullet list at the end of the first section.
2. The authors should also provide the number of labels within the final dataset.
3. The authors should improve Figure 3 by enlarging it.
4. Please also provide more explanatory captions.
5. Did the authors test further attention mechanisms? Several recent works have already evaluated the impact of attention within YOLO networks (e.g., https://doi.org/10.3390/rs17061027 or https://doi.org/10.1016/j.atech.2025.101324). Please provide further details accordingly.
6. Please highlight the best results in bold font.
7. I could not find Table 2. Please provide further details on hyperparameter selection.
8. Figure 7 provides low value and can be removed.
9. Please provide a higher resolution version of Figure 9.
10. Please provide the densities in Table 5.
11. The authors should consider providing the code for reproducibility and open science practices.

Author Response

1. The authors should provide the innovation as a bullet list at the end of the first section.

We appreciate the reviewer's suggestions and have presented the innovative points in a bulleted list at the end of the first section (Introduction). The specific content is as follows: “The main contributions of this study are as follows: "1. An improved YOLOv12 algorithm is proposed, specifically optimized to address the challenges posed by melon pests, such as their minute size and inconsistent dimensions. This enhancement achieves higher detection accuracy while maintaining a low number of parameters and computational requirements. Additionally, the refined model effectively detects pests across various crops, demonstrating excellent adaptability. 2. The C3k2-B module is designed by introducing an adaptive residual block (ARBlock) to replace the original block module in the C3k2 architecture. This transformation creates a lightweight and efficient feature extraction module, enabling the model to capture features across different receptive fields. Furthermore, the fusion layer between the Neck and Head networks is modified to enhance the model's utilization of shallow-layer features.”(Page 3, Lines 122-132)

2. The authors should also provide the number of labels within the final dataset.

We appreciate the reviewer's suggestion and have added label information for the training and validation sets (Figure 3).(Page 4, Line 162)

3. The authors should improve Figure 3 by enlarging it.

We appreciate the reviewers' suggestions. We have enlarged Figure 3 (now Figure 4, Page 5, Line 179) as a whole and increased the font size within the figure. Figures 6(Page 7, Line 246) and 7(Page 9, Line 334) have been replaced accordingly.

4. Please also provide more explanatory captions.

We have revised the titles of Sections 2.2( YOLOv12 network improvements, Page 5, Line 164), 2.2.2(Improved Neck and Head network fusion strategy, Page 7, Line 222), 3.2(Model performance evaluation metrics, Page 10, Line 343), and 3.4( Performance comparison between C3k2 and C3k2-B Modules, Page 13, Line 441) based on the reviewers' suggestions.

5. Did the authors test further attention mechanisms? Several recent works have already evaluated the impact of attention within YOLO networks (e.g., https://doi.org/10.3390/rs17061027 or https://doi.org/10.1016/j.atech.2025.101324). Please provide further details accordingly.

Thank you for the reviewer's questions. The paper has now conducted comparative experiments on the collaborative performance of EMA, CA, SimAM, ECA, CBAM, and SE attention mechanisms. We reviewed the paper https://doi.org/10.1016/j.atech. 2025.101324 “Incremental Learning with Domain Adaptation for Tomato Plant Phenotyping.” The paper focuses on the Convolutional Block Attention Module (CBAM) and Shuffle Attention (SA) as its core attention mechanisms, systematically validating their performance and applicability in tomato plant phenotyping tasks through experimental verification. In our work, we also employed CBAM for comparative experiments. Therefore, we added a set of experiments using the SA attention mechanism to the attention comparison experiments in Section 3.6, which are presented in Table 6.(Page 16, Line 548) In the paper “AG-Yolo: Attention-Guided Yolo for Efficient Remote Sensing Oriented Object Detection” (https://doi.org/10.3390/rs17061027), the authors propose an Attention Branch—a supervised spatial attention mechanism tailored for remote sensing image-oriented object detection. However, the paper does not provide access to the specific code. We will reproduce the results based on the architecture outlined in the paper in the future.

6. Please highlight the best results in bold font.

We thank the reviewers for their suggestions and have highlighted the best results in bold throughout all experimental tables. Tabel 3(Page 11, Line 418), Tabel 4(Page 13, Line 437), Tabel 5(Page 14, Line 483), Tabel 6(Page 16, Line 548), Tabel 8(Page 18, Line 581), Tabel 9(Page 18, Line 583).

7. I could not find Table 2. Please provide further details on hyperparameter selection.

We appreciate the reviewer's suggestion and apologize for the omission of Table 2 in the paper. Table 2(Page 10, Line 342) has now been added to Section 3.1, providing details on hyperparameter selection.

8. Figure 7 provides low value and can be removed.

We appreciate the reviewer's suggestions. Our Figure 7 (now Figure 8, Page 12, Line 425) is a line chart of the model training results, which provides valuable information. Following another reviewer's recommendation, we have redrawn Figure 7 (now Figure 8) with enhanced resolution. We have enlarged the metric curves for the final 10 training rounds and added specific numerical values for all four metrics of our improved model (mAP50, P, R, and mAP50-95).

9. Please provide a higher resolution version of Figure 9.

We appreciate the reviewer's suggestion and have provided a higher-resolution Figure 9 (now Figure 10, Page 14, Line 455).

10. Please provide the densities in Table 5.

We appreciate the reviewer's suggestion. We understand that “density” refers to “model density,” so we have updated the model names in Table 5 to include the corresponding model density, such as YOLOv12n.(Page 14, Line 483)

11. The authors should consider providing the code for reproducibility and open science practices.

Thank you for your valuable feedback and for recognizing the importance of our work. We fully understand the need to provide code that enables reproducibility and open science practices. However, due to our laboratory's policy, we are unable to release the source code publicly prior to the publication of this study. However, we remain committed to assisting others in replicating experiments. If you or other researchers have specific questions or require further details, please feel free to contact us. We will respond promptly with a detailed explanation. Thank you for your understanding, and we look forward to contributing to the research community in this manner.

Round 2

Reviewer 2 Report

Comments and Suggestions for Authors

I have no further questions, I recommend this paper for publication.

Author Response

Thank you to the reviewer for your approval.

Reviewer 3 Report

Comments and Suggestions for Authors

No more comments.

Author Response

(The authors gave the same response as above.)

Reviewer 4 Report

Comments and Suggestions for Authors

The authors successfully fixed the issues highlighted during the previous round of review. I strongly suggest the paper for acceptance, given its significance and value for practictioneers.

Author Response

(The authors gave the same response as above.)
